# A chemical-genetic system to rapidly inhibit the PP2A-B56 phosphatase reveals a role at metaphase kinetochores

Lindsey A. Allan [1,4], Andrea Corno [1,4], Juan Manuel Valverde [1,4], Rachel Toth [2], Tony Ly [3] & Adrian T. Saurin [1]✉

Serine-threonine phosphatases have been challenging to study because of the lack of specific inhibitors. Their catalytic domains are druggable, but these are shared or very similar between individual phosphatase complexes, precluding their specific inhibition. Instead, phosphatase complexes often achieve specificity by interacting with short linear motifs (SLiMs) in substrates or their binding partners. We develop here a chemical-genetic system to rapidly inhibit these interactions within the PP2A-B56 family. Drug-inducible recruitment of ectopic SLiMs ("directSLiMs") is used to rapidly block the SLiM-binding pocket on the B56 regulatory subunit, thereby displacing endogenous interactors and inhibiting PP2A-B56 activity within seconds. We use this system to characterise PP2A-B56 substrates during mitosis and to identify a role for PP2A-B56 in allowing metaphase kinetochores to properly sense tension and maintain microtubule attachments. The directSLiMs approach can be used to inhibit any other phosphatase, enzyme or protein that uses a critical SLiM-binding interface, providing a powerful strategy to inhibit and characterise proteins once considered "undruggable".

The PPP family of serine-threonine phosphatases dephosphorylate the majority of phospho-Ser/Thr residues in cells[1,2]. This large family of enzymes have been notoriously difficult to study because they share the same or very similar catalytic subunits, precluding their specific targeting by small molecule inhibitors[3]. Natural toxins have been identified that display some preference for either the PP1 or PP2A catalytic domains[4], but these domains are still shared between multiple different holoenzyme complexes. For example, PP2A exists as a heterotrimeric phosphatase containing a PP2A catalytic domain, a scaffold domain, and one of four different classes of divergent regulatory subunits - B' (B55, PR55), B" (B56, PR61), B'" (PR48/PR70/PR130), and B"" (PR93/PR110 or Striatins)[5]. It is protein-protein interactions mediated by these regulatory subunits that target PP2A to specific substrates. Each regulatory subunit is present as multiple different

isoform variants, further increasing diversity within the PP2A family. PP2A-B56 holoenzymes, for example, use six different B56 isoforms encoded by separate genes[6].

A major substrate binding pocket on B56, which is well conserved in all B56 isoforms, binds to short linear motifs (SLiMs) containing a consensus LxxIxE sequence[7]. A wide variety of PP2A-B56 substrates contain LxxIxE motifs, and their presence within conserved unstructured regions of proteins can be used to predict PP2A-B56 substrates[7,8]. LxxIxE motifs often contain a serine or threonine residue within or immediately after this motif, and phosphorylation of these residues can enhance B56 binding strength[9–12]. This allows phosphorylation inputs to recruit PP2A-B56 to specific substrates or locations at certain times. For example, CDK1 and PLK1-dependent binding of PP2A-B56 to BUBR1 allows this interaction to occur specifically at kinetochores

[1]Division of Cancer Research, Jacqui Wood Cancer Centre, School of Medicine, University of Dundee, Dundee DD1 9SY, UK. [2]MRC Protein Phosphorylation and Ubiquitylation Unit Reagents and Services Laboratory, School of Life Sciences, University of Dundee, Dundee, UK. [3]Molecular Cell and Developmental Biology, School of Life Sciences, University of Dundee, Dundee DD1 5EH, UK. [4]These authors contributed equally: Lindsey A. Allan, Andrea Corno, Juan Manuel Valverde. ✉e-mail: a.saurin@dundee.ac.uk

during mitosis, where it is needed to promote dephosphorylation of sites that would otherwise impede microtubule binding[13,14]. This is important to stabilise kinetochore-microtubule attachments and allow proper chromosome segregation.

Mutation of specific LxxIxE motifs within substrates has been crucial for characterising PP2A-B56 function. For example, mutation of the LxxIxE motif in BUBR1 uncovered roles for PP2A-B56 in promoting initial kinetochore-microtubule attachments and antagonising the spindle assembly checkpoint[9,10,15–22]. Similar strategies have been used to identify roles for PP2A-B56 in cytokinesis via RacGAP1 binding[7], and homologous recombination via BRCA2 binding[23].

A complementary approach to characterise PP2A-B56 binding partners and substrates is to mutate the LxxIxE-binding pocket on B56 (hereafter referred to as the substrate binding pocket). Mutations of key residues in B56 can prevent LxxIxE motif binding, and this was used to identify a role for this substrate binding pocket in promoting Sgo1-binding to protect cohesion during mitosis[24]. This type of approach can also be used to assess PP2A-B56 function more globally, for example to look for binding partners and phosphorylation events that change when the substrate pocket is mutated. However, it takes days to switch to a mutant form of B56, during which time cells accumulate in mitosis with unattached kinetochores due to lack of BUBR1 binding. This complicates the assessment of PP2A-B56 function during later mitosis or at any other cell cycle stage.

An alternative approach was developed by the Nilsson lab to block the substrate binding pocket by the inducible expression of a high-affinity tetrameric LxxIxE peptide[25]. This was shown to bind and occlude the substrate binding pocket on B56, allowing the identification of PP2A-B56 substrates during mitosis or S-phase[25]. However, it still takes 12 hours to express such a protein to high enough levels to inhibit B56 function, complicating interpretations about whether the resulting phospho-proteomic changes are direct or indirect. A method to more rapidly inhibit the substrate binding pocket on B56 would allow acute inhibition of enzyme activity during any cell cycle stage or any process, thus facilitating the analysis of PP2A-B56 function.

Here we develop a chemical-genetic system to inducibly recruit an LxxIxE-containing peptide to block the B56 substrate-binding pocket within seconds of small molecule addition. This can rapidly compete off B56 substrates and inhibit PP2A-B56 function, as evidenced by the fact that increased substrate phosphorylation is observed in as little as 15 seconds. We use this to comprehensively characterise acute phosphorylation changes proteome-wide following PP2A-B56 inhibition during mitosis, and to identify a role for PP2A-B56 in antagonising Aurora B and maintaining stable kinetochore-microtubule attachments at metaphase. We name this approach directSLiMs - for **d**rug-**i**nducible **r**ecruitment of **ect**opic **SLiMs** - because it is a simple yet powerful approach to block critical SLiM-based interactions on other phosphatases[3], enzymes[26] or even non-enzymatic proteins[26,27].

## Results

### Developing a directSLiMs method to rapidly inhibit PP2A-B56

We set out to develop a rapid drug-inducible recruitment strategy to block the LxxIxE binding pocket on B56 and thereby inhibit substrate dephosphorylation (Fig. 1A). We chose to use the FKBP-FRB[T2098L] interaction that can be induced by rapamycin or rapamycin analogues that do not bind the TORC1 complex (rapalogs)[28], to recruit a non-phospho-dependent LxxIxE peptide, initially characterised from Ebola (**L**PTI**H**E**E**EEE: hereafter named LIE1), to the PP2A-R1A scaffolding subunit[8]. We used a ribosome-skipping T2A sequence to produce separate LIE1[FRB] and [FKBP]R1A proteins from the same doxycycline-inducible plasmid that was integrated into a FRT site within HeLa-FRT cells (Fig. 1B). Immunoprecipitations of two different B56 subunits, showed that this plasmid can be used to fully replace endogenous R1A subunit with [FKBP]R1A in PP2A-B56 complexes (Supplementary Fig. 1A). Furthermore, anti-FLAG immunoprecipitations showed that the FLAG-tagged [FKBP]R1A subunit is able to bind to the PP2A-catalytic domain and various B56 regulatory subunits, indicating stable association into PP2A-B56 holoenzyme complexes (Fig. 1C: left panel). Rapamycin treatment for 30 mins, to induce recruitment of the co-expressed LIE1[FRB], did not perturb PP2A-B56 holoenzyme assembly, but it did efficiently compete off B56-binding partners that are known to interact via the LxxIxE binding pocket (Fig. 1C: right panel; GEF-H1, BubR1, RepoMan)[7].

PP2A-B56 is recruited to kinetochores during mitosis by binding to an LxxIxE motif in the kinetochore attachment regulatory domain (KARD) of BUBR1 (amino acids 664–681)[9,10]. From this position it regulates many kinetochore phosphorylation events, including the adjacent BUBR1-pT620, which is a phospho-dependent recruitment site for Polo-like Kinase 1 (PLK1) (Fig. 1D)[17,18,29,30]. A 20-min treatment with rapamycin was sufficient to displace [FKBP]R1A from the kinetochore and to increase BUBR1-T620 phosphorylation in LIE1[FRB] cells (Fig. 1E-F and Supplementary Fig. 1B-C). This required the LIE1 peptide to bind the substrate binding pocket on B56, because rapamycin had no effect in [FKBP]R1A cells expressing an AAA[FRB] mutant that is analogous to LIE1[FRB] but contains alanine residues in key B56 binding positions (Fig. 1E-F and Supplementary Fig. 1B-C). In summary, drug-inducible recruitment of the ectopic SLiM LIE1 (directSLiM[LIE1]) to the R1A subunit competes off B56 substrates and displaces PP2A-B56 from the kinetochore, leading to increased phosphorylation of the well-established kinetochore substrate BUBR1.

To examine how quickly directSLiM[LIE1] could inhibit PP2A-B56 we performed similar analysis after 0–30 mins of rapamycin treatment. This demonstrated that PP2A delocalisation and enhanced BUBR1 phosphorylation occurred within 1–2 minutes of rapamycin addition (Fig. 1G). We also used a rapamycin analogue (also known as A/C heterodimeriser, but hereafter called rapalog) that binds to the FRB-T2098L mutant present in the LIE1[FRB], but not to endogenous FRB, thus preventing inhibition of the endogenous TORC1 complex[31]. Rapalog treatment similarly inhibited PP2A-B56, although the timescale of inhibition was slightly delayed (by 5–10 mins), perhaps due to differences in FRB-FKBP affinity and/or rapalog cell permeability (Fig. 1H).

### Using directSLiMs to identify PP2A-B56 substrates during mitosis

To globally assess phosphorylation changes following PP2A-B56 inhibition, nocodazole-arrested mitotic directSLiM[LIE1] cells were treated for 30 min with DMSO, rapamycin or rapalog, followed by TMT labelling, phosphopeptide enrichment and measurement by mass spectrometry (MS) (Fig. 2A). A total of 9907 phospho-sites were quantified and of these 187 exhibited a significant >1.5-fold increase following rapamycin or rapalog treatment (Fig. 2B, Supplementary Fig. 2A and Supplementary Data 1).

There was good correlation between the different heterodimerisers in the directSLiM[LIE1] expressing cell lines (Fig. 2C, Supplementary Fig. 2B), and similar changes were not observed in parental HeLa-FRT cells (Fig. 2B-C), implying they were due to PP2A-B56 inhibition. In fact, rapamycin and rapalog alone in HeLa-FRT cells cause very few phosphorylation changes after 30 min treatment (Fig. 2B). The lack of effect with rapalog is expected because it cannot bind endogenous TORC1, however, the lack of effect of rapamycin might be explained by the fact that the TORC1 complex is inhibited during mitosis[32,33]. The proteins displaying increased phosphorylation following drug treatment in directSLiM[LIE1] cells were likely direct substrates of PP2A-B56, since we detected specific enrichment of validated (adj.p = 0.004, OR = 5.1) and predicted (adj.p = 0.03, OR = 1.5) LxxIxE motifs within these proteins (Supplementary Fig. 2C), and the most significant hits were on the well-characterised substrate BUBR1 (Fig. 2B, C). The increasing phospho-sites were enriched for Aurora and PLK1 consensus motifs and depleted for proline-directed CDK motifs when compared to the

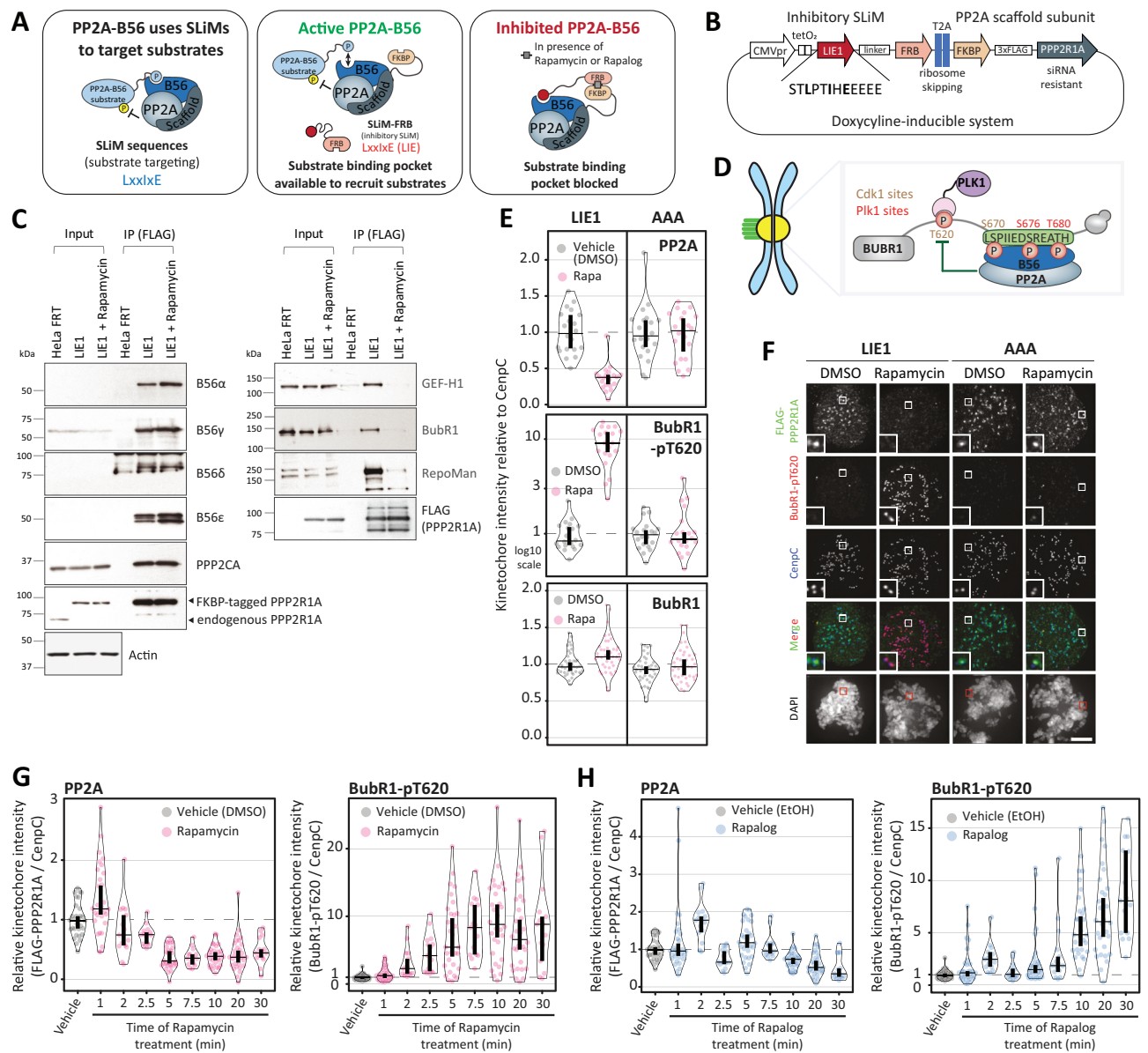

**Fig. 1 | A chemical-genetic system to rapidly inhibit PP2A-B56. A** Schematic illustrating the principle of the drug-inducible recruitment of ectopic SLiMs (directSLiMs) system. **B** Representation of the plasmid used to implement the directSLiMs strategy in HeLa-FRT cells. **C** Immunoblot of the PP2A complex (left column) and PP2A substrates (right column), following FLAG immunoprecipitation from nocodazole-arrested HeLa-FRT cells with/without directSLiM[LIE1] expression, -/+ rapamycin for 30 min. Representative of two experiments. **D** Schematic illustrating how PP2A-B56 regulates the phosphorylation of the T620 site on BUBR1. **E–F** Quantifications (**E**) and representative example immunofluorescence images (**F**) to show the effects of PP2A-B56 inhibition on the levels of FLAG-PPP2R1A, BUBR1-pT620 and BUBR1 at unattached kinetochores in nocodazole-arrested HeLa-FRT cells expressing the directSLiMs[LIE1/AAA] and treated with vehicle or rapamycin

for 20 min. Kinetochore intensities from 20–40 cells, 2–4 experiments. The insets show magnifications of the outlined regions. Scale bars: 5 μm. Inset size: 1.5μm **G–H**. Effects of PP2A-B56 inhibition on the levels of FLAG-PPP2R1A and BUBR1-pT620 at unattached kinetochores, in nocodazole-arrested HeLa-FRT cells expressing directSLiM[LIE1] and treated with rapamycin (**G**) or rapalog (**H**) for 0-30 mins. Kinetochore intensities from 15–30 cells, 1–2 experiments. Source data are provided as a Source Data file. Data information: Kinetochore intensities in E–H are normalized to directSLiM[LIE1] vehicle condition. Violin plots show the distributions of kinetochore intensities between cells. For each violin plot, each dot represents an individual cell, the horizontal line represents the median and the vertical one the 95% CI of the median, which can be used for statistical comparison of different conditions (see Methods).

---

non-changing phospho-sites (Fig. 2D–E). This is consistent with previous data showing that PP2A-B56 displays reduced activity against CDK substrates[25], and is mainly involved in antagonising PLK1[17,18], MPS1[21] and Aurora B activities[15,34] (note that MPS1 has a very similar substrate preferences to PLK1[35,36]).

To validate if the increase in B56 substrate phosphorylation can be observed using a complementary approach, we used antibodies against a wide range of phosphorylation sites on BUBR1 interacting proteins. BUBR1 binds directly to BUB1, and the BUB1:BUBR1 complex

is recruited to kinetochores by binding to phosphorylated MELT motifs on KNL1[13]. The schematic in Fig. 3A shows the complex along with relevant phospho-sites for which validated phospho-antibodies are available. Importantly, all of these phosphorylation sites are known to be regulated directly or indirectly by PP2A-B56[15,17,18,20,37]. Immunofluorescence analysis demonstrated the expected phosphorylation increases following rapamycin treatment in directSLiM[LIE1] cells, implying localised inhibition of B56 at the kinetochore (Fig. 3B and Supplementary Fig. 3). However, we also noticed an increase in basal

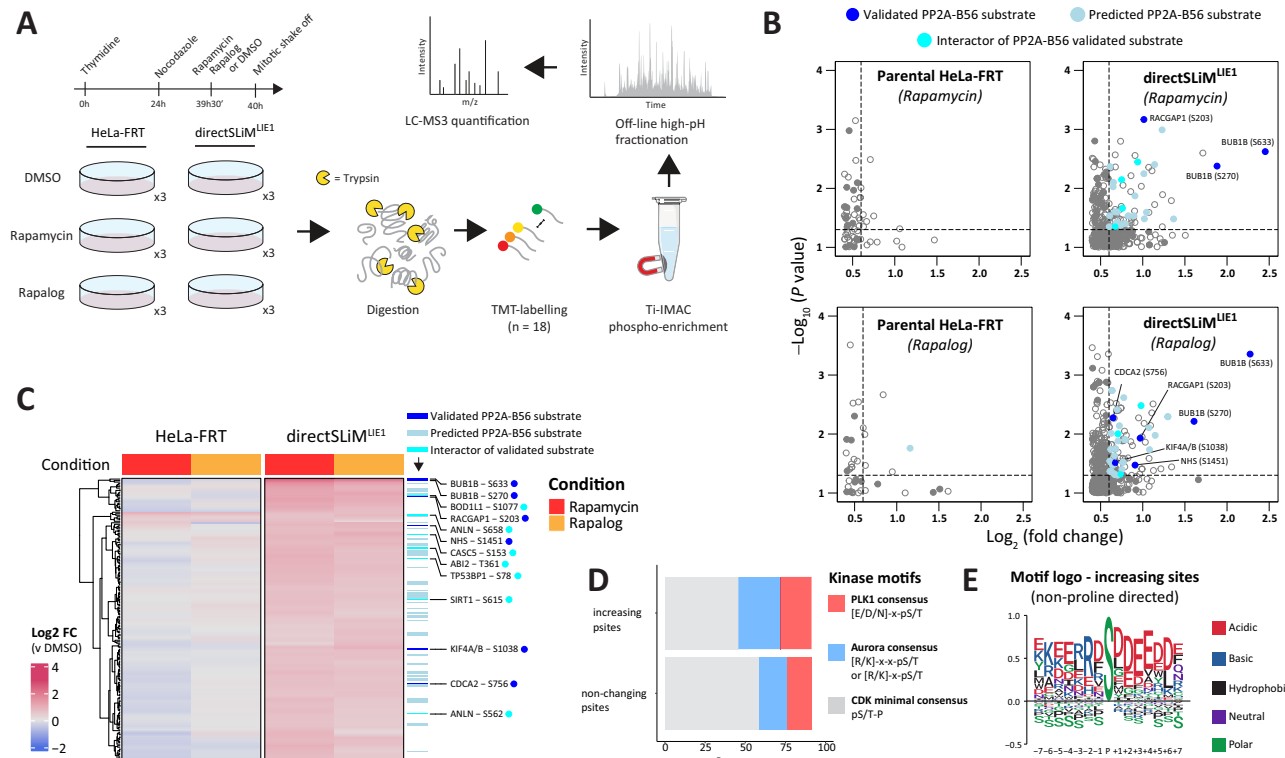

**Fig. 2 | Global protein phosphorylation changes after PP2A-B56 inhibition in directSLiM^LIE1 cells. A** Experimental design: mitotically arrested directSLiM^LIE1 and HeLa-FRT cells were treated with rapamycin, rapalog or vehicle before harvest ($n = 3$ for each condition). Following cell lysis, proteins were digested, and peptides were TMT-labeled. Next, phosphopeptides were enriched with Ti-IMAC magnetic beads, followed by high-pH reversed-phase fractionation and measurement by LC-MS. **B** Zoomed-in right upper quadrant of volcano plot showing upregulated phosphorylation sites after treatment with rapamycin or rapalog. Solid dots are either validated substrates, predicted substrates or interactors of validated PP2A-B56 substrates (see Methods for details). Only hits above a -Log₁₀(*P* value = 0.05)

(Two-sided Student's t-test) and >1.5-fold change are color coded. **C** Heat map of phosphorylation sites upregulated after PP2A-B56 inhibition in directSLiM^LIE1 cells. Each row represents the intensity of a phosphorylation site ($n = 187$). Intensities are normalised to vehicle-treated condition for each cell line to show the relative fold-change upon rapamycin/rapalog addition. Gene names are added for all phospho-sites from validated substrates or interactors of validated substrates. **D** Percentage of Aurora kinase, PLK1 and CDK consensus motifs in the increasing and non-changing phosphorylation sites. **E** Sequence logo of non-proline directed upregulated phosphorylation sites normalized against background (all non-changing phosphosites used as background). Source data are provided as a Source Data file.

phosphorylation of these sites in the absence of rapamycin treatment, compared to untreated HeLa-FRT controls.

We were initially confused as to why BUBR1-pT620, which is the closest to the B56 binding site (Fig. 3A), did not also increase basally in directSLiM^LIE1 cells (Fig. 1E–H). However, this analysis was performed with a 1-min pre-extraction procedure to reduce background cytoplasmic staining and allow visualisation of R1A localisation to kinetochores. Repeat analysis of BUBR1-pT620 without pre-extraction showed increased basal phosphorylation in the absence of rapamycin treatment in directSLiM^LIE1 cells (Fig. 3C and Supplementary Fig. 3). The pre-extraction may have also altered the apparent kinetics of inhibition because BUBR1-pT620 was observed to increase within 15 seconds of rapamycin-treatment without pre-extracting cells (Fig. 3D).

We next wanted to assess whether basal inhibition was also apparent in our proteomic analysis. It is possible that substrates displaying increased basal phosphorylation in directSLiM^LIE1 cells could be missed using a 1.5-fold cutoff upon rapamycin/rapalog treatment. Therefore, we filtered based on a >1.5-fold increase from DMSO-treated HeLa-FRT cells instead, and applied a less stringent requirement for a significant increase from DMSO-treated directSLiM^LIE1 cells. This reanalysis showed that the sites that increased upon rapamycin/rapalog treatment in directSLiM^LIE1 cells had a tendency to have a higher level of basal phosphorylation compared to HeLa-FRT cells (Fig. 3E), consistent with the basal inhibition observed in our immunofluorescence analysis. Together, this data implies that LIE1-B56 affinity is sufficiently strong to compete off some B56 interactors even

in the absence of rapamycin. Therefore, we set out to generate LIE sequences that had reduced affinity for B56, with the expectation that these should still be inhibitory when tethered to PP2A by hetero-dimeriser treatment.

**Evolving directSLiMs to reduce basal PP2A-B56 inhibition**

We first used a range of LxxIxE peptides that had published Kd's for B56 ranging from 750 nM to 110 µM, which was significantly higher than the 80 nM Kd of LIE1 (Fig. 4A)[7,8,12]. The strongest binding peptide in these studies (LIE2) behaved similarly to LIE1, as assessed by BUBR1-pT620 immunofluorescence or live-cell imaging experiments to quantify chromosome alignment phenotypes that result from decreased BUBR1-B56 interaction (Fig. 4B and Supplementary Fig. 4A). Peptides with lower affinities than LIE2 (LIE3-7) exhibited reduced inhibition both in the absence or presence of rapamycin. Therefore, we next tried to remove key acidic residues C-terminal to the LxxIxE motif in the original LIE1 peptide, which is predicted to progressively decrease binding strength[7,11,12], creating eight further LIE peptides (LIE8-14; Fig. 4C). This identified that the sequence LIE9 was able to achieve strong B56 inhibition upon rapamycin treatment, but with minimal effects on chromosome alignment or BUBR1-T620 phosphorylation in the absence of rapamycin (Fig. 4D and Supplementary Fig. 4B). DirectSLiM^LIE10 cells also had low basal effects and good inducible inhibition, however mitotic duration was still slightly extended compared to directSLiM^AAA controls (Supplementary Fig. 4C), which may be due to subtle basal inhibition. DirectSLiM^LIE9

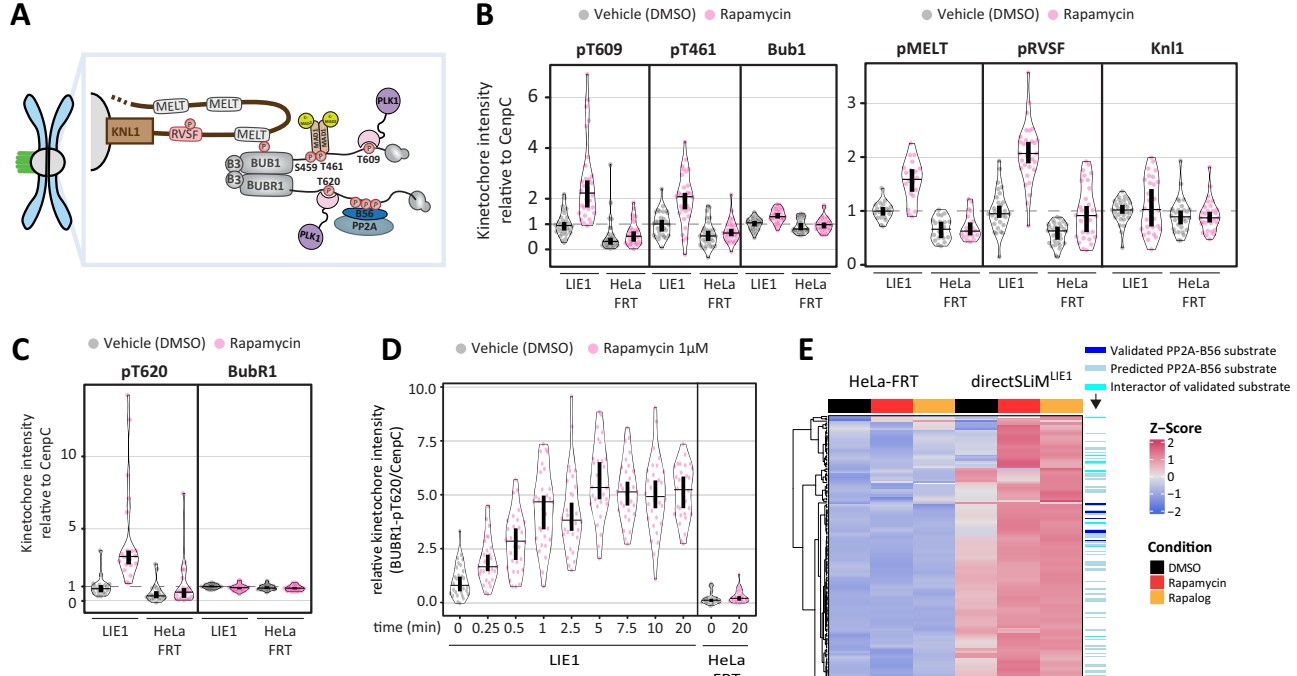

**Fig. 3 | Basal inhibition of PP2A-B56 is observed in directSLiM$^{LIE1}$ cells.**
**A** Schematic illustrating kinetochore phosphosites on the BUB complex and KNL1 that are regulated by PP2A-B56. **B–C** Effects of PP2A-B56 inhibition on the levels of BUB1-pT609, BUB1-pT461 and BUB1 (**B**, left) or KNL1-pMELT, KNL1-pRVSF and KNL1 (**B**, right) or BUBR1-pT620 and BUBR1 (**C**) at unattached kinetochores, in nocodazole-arrested HeLa-FRT cells with/without directSLiM$^{LIE1}$ expression and treated with vehicle or rapamycin for 20 min. Kinetochore intensities from 20–30 cells, 2–3 experiments. **D** Levels of BubR1-pT620 at unattached kinetochores in nocodazole-arrested directSLiM$^{LIE1}$ cells treated with rapamycin for indicated times.

Kinetochore intensities from 30 cells, 3 experiments **E** Heat map of phosphorylation sites upregulated after PP2A-B56 inhibition in directSLiM$^{LIE1}$ cells. Each row shows the z-scored intensity of a phosphorylation site ($n = 170$). Source data are provided as a Source Data file. Data information: Kinetochore intensities in **B–C** are normalized to directSLiM$^{LIE1}$ vehicle condition. Violin plots show the distributions of kinetochore intensities between cells. For each violin plot, each dot represents an individual cell, the horizontal line represents the median and the vertical one the 95% CI of the median, which can be used for statistical comparison of different conditions (see Methods).

---

cells also consistently produce no or negligible basal effects against other sites on BUB1 and KNL1 (Fig. 4E and Supplementary Fig. 4D).

## Proteomic analysis using an optimised directSLiM to identify PP2A-B56 substrates during mitosis

We next performed a proteomic analysis in directSLiM$^{LIE9}$ cells to identify phosphorylation changes following rapamycin or rapalog treatment. Considering rapamycin or rapalog treatment induced minimal phospho-changes in HeLa-FRT cells (Fig. 2B–C), we decided to use directSLiM$^{AAA}$ cells instead as the control condition this time, to examine if heterodimeriser-induced recruitment of FRB to PP2A-R1A impacted on PP2A function, for example, due to steric effects. We also used a modified phospho-peptide enrichment method that enhanced phospho-peptide recovery (Fig. 5A, Supplementary Fig. 5A). Using this method, we identified a total of 15,518 phospho-sites in mitotically arrested directSLiM$^{LIE9}$ cells, of which 149 exhibited a significant >1.5-fold change following rapamycin or rapalog treatment (Supplementary Data 2). Some of the most significantly increasing phospho-sites were in well characterised PP2A-B56 substrates, including BUBR1 (BUB1B), RepoMan (CDCA2), RACGAP1, KIF4A and NHS (Fig. 5B-C, Supplementary Fig. 5B). There was also a strong significant enrichment for validated PP2A-B56 binding motifs in the proteins that contained increasing phospho-sites (adj.$p = 3.1 \times 10^{-9}$, OR = 12.5; Supplementary Fig. 5C). We observed enrichment of potential Aurora kinase and PLK sites, and depletion of potential CDK sites (Fig. 5D-E), as also observed in the previous MS analysis with directSLiM$^{LIE1}$ cells (Fig. 2D-E).

Basal phosphorylation levels were also generally lower in directSLiM$^{LIE9}$ cells, when compared to the directSLiM$^{LIE1}$ cells, using the same less stringent filtering we used to capture any basally high phosphorylation events (Fig. 5F; compare to Fig. 3E). This indicates

that LIE9-dependent PP2A-B56 inhibition is largely dependent on heterodimeriser treatment. We did observe more heterodimeriser-dependent changes in directSLiM$^{AAA}$ cells than we had observed previously in parental HeLa-FRT cells (compared Figs. 5B and 2B), but these were generally not the phospho-sites that increased further upon heterodimeriser treatment (Fig. 5F). These increased phosphorylation sites could reflect steric effects of AAA$^{FRB}$ recruitment to PP2A complexes. The correlation between the effects of rapamycin or rapalog was good, but not as high when compared to the directSLiM$^{LIE1}$ cells (Fig. 5C, Supplementary Fig. 5D). This may be due to the lower affinity of the LIE9 sequence slowing the rate of PP2A-B56 inhibition, which could disproportionately affect rapalog-treated cells which were generally inhibited slower (Fig. 1G–H). Importantly however, all increases to validated substrates, or to proteins that interact with validated substrates, increase following both rapamycin and rapalog treatment (Fig. 5F). This suggests that the subset of targets that change with both drugs may be the most reliable.

## Using directSLiMs to characterise the role of PP2A-B56 at metaphase kinetochores

The ability to rapidly inhibit PP2A-B56 within seconds (Fig. 1G–H) creates opportunities to examine PP2A function in cell cycle phases that last only minutes. One good example is metaphase, which is the phase when all kinetochores have attached to microtubules. Metaphase lasts only 5–10 minutes but during that time the SAC signal is switched off, the mitotic checkpoint complex is disassembled, and active APC/C$^{CDC20}$ degrades its key mitotic substrates cyclin B and securin, allowing cells to progress into anaphase and exit mitosis[38]. If kinetochores become detached during metaphase then the SAC is reimposed and mitotic progression is rapidly halted[39]. PP2A-B56 is

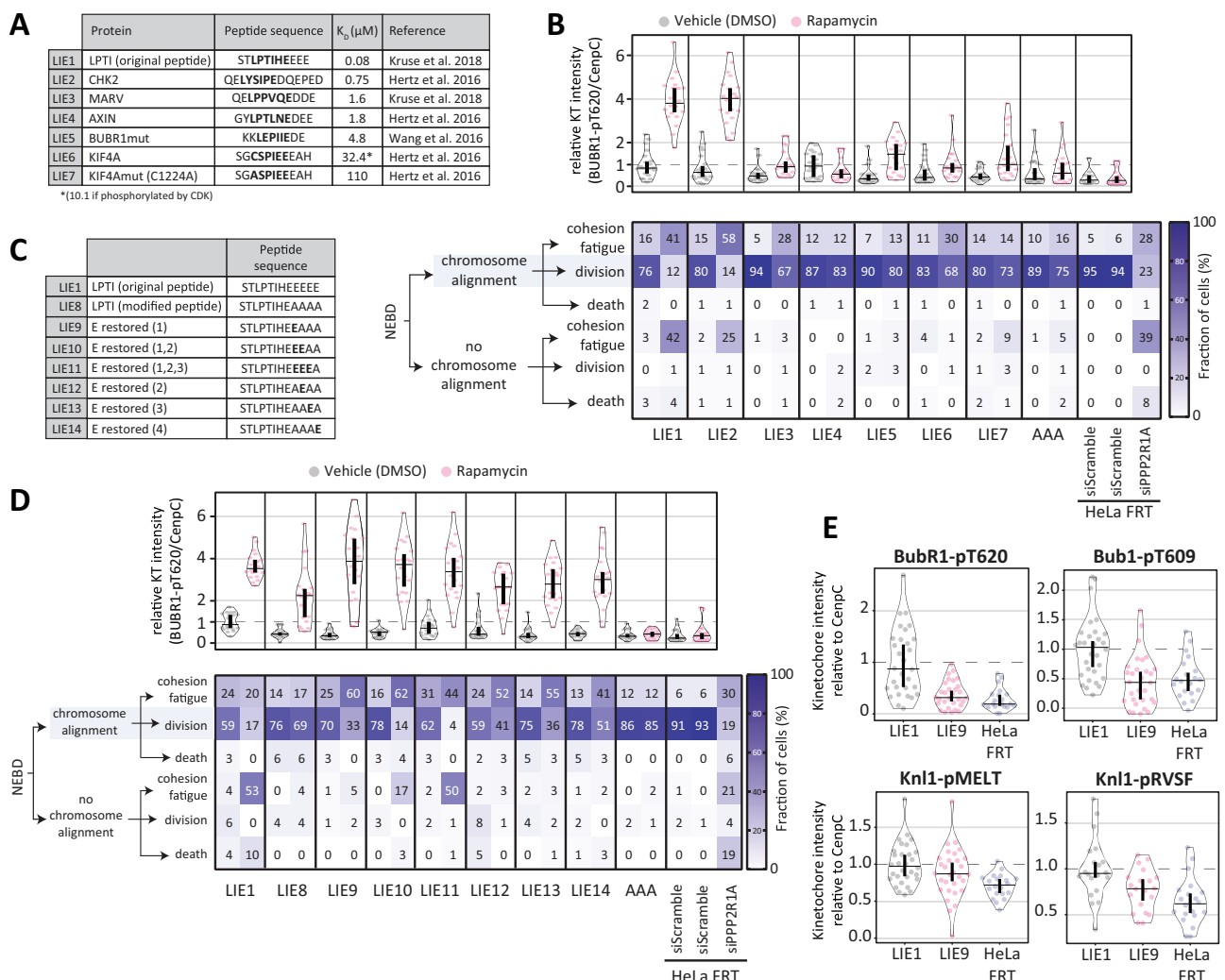

**Fig. 4 | Optimization of the directSLiMs strategy to reduce basal inhibition of PP2A-B56.** **A** Table reporting the tested SLiMs with a lower binding affinity for PP2A-B56 in comparison to LIE1. For each peptide sequence, the SLiM is highlighted in bold. **B** Effects of PP2A-B56 inhibition on the levels of BUBR1-pT620 at unattached kinetochores (top) and on the mitotic cell fate after nuclear envelope breakdown (NEBD, bottom) in HeLa-FRT cells with/without directSLiMs[LIE1-7/AAA] expression and treated with vehicle or rapamycin. Top graph: Kinetochore intensities from 20 nocodazole-arrested cells, 2 experiments except HeLa-FRT, DMSO: 10 cells. The treatment with vehicle/rapamycin was performed for 20 min prior fixation. Bottom graph: heatmap showing the mean frequencies of cell fate after NEBD in each condition: 2 experiments, 17–50 cells per condition per experiment (see also Supplementary Fig. 4A). **C** Table reporting the tested SLiMs with different numbers of acidic residues C-terminal to the LxxIxE motif. The acidic residues restored from LPTI (modified peptide) are highlighted in bold. **D** Effects of PP2A-B56 inhibition on the levels of BUBR1-pT620 at unattached kinetochores (top) and on the mitotic cell fate after nuclear envelope breakdown (NEBD, bottom) in HeLa-

FRT cells with/without directSLiMs[LIE1/LIE8-14/AAA] expression and treated with vehicle or rapamycin. Top graph: Kinetochore intensities from 20 nocodazole-arrested cells, 2 experiments. The treatment with vehicle/rapamycin was performed for 20 min prior fixation. Bottom graph: heatmap showing the mean frequencies of cell fate after NEBD in each condition: 2 experiments, 42-50 cells per condition per experiment (see also Supplementary Fig. 4B). **E** Levels of BubR1-pT620, Bub1-pT609, Knl1-pMELT and Knl1-pRVSF at unattached kinetochores, in nocodazole-arrested HeLa-FRT cells expressing directSLiMs[LIE1/LIE9] and treated with vehicle for 20 min. Note that the distributions of directSLiM[LIE1] and directSliM[LIE9] are also shown in Supplementary Fig. 4D. Kinetochore intensities from 20 to 30 cells, 2–3 experiments. Source data are provided as a Source Data file. Data information: Kinetochore intensities in **B, D** and **E** are normalized to directSLiM[LIE1] vehicle condition. Violin plots show the distributions of kinetochore intensities between cells. For each violin plot, each dot represents an individual cell, the horizontal line represents the median and the vertical one the 95% CI of the median, which can be used for statistical comparison of different conditions (see Methods).

required to stably attach kinetochores to microtubules during prometaphase[9,10,40], but whether PP2A-B56 is still required to maintain those attachments on aligned chromosomes at metaphase remains unknown (Fig. 6A). Fixed chromosome alignment assays in metaphase-arrested cells demonstrated that 30 min rapamycin treatment was sufficient to detach chromosomes from microtubules in directSLiM[LIE9] cells, but not in directSLiM[AAA] cells, which is consistent with reduced PP2A-B56 recruitment to BUBR1 inhibiting kinetochore-microtubule attachments (Fig. 6B). However, PP2A-B56 also binds Sgo1 at the centromere to maintain centromeric cohesion, in a manner that is dependent on its LxxIxE-binding pocket[24]. To test whether loss of

centromeric cohesion contributes to the chromosome misalignment phenotypes at metaphase, we depleted WAPL because this can preserve cohesion in the absence of PP2A[41,42]. WAPL depletion was able to fully rescue misaligned chromosomes following Sgo1 depletion, as expected[41,42], but it did not affect the detachment of chromosomes in directSLiM[LIE9] cells treated with rapamycin (Fig. 6B). Therefore, B56 inhibition at metaphase impairs chromosome alignment, most likely by dissociating PP2A from the outer kinetochore BUBR1 to detach kinetochore-microtubules.

To determine the kinetics of kinetochore-microtubule detachment at metaphase, we used live assays to quantify the time to

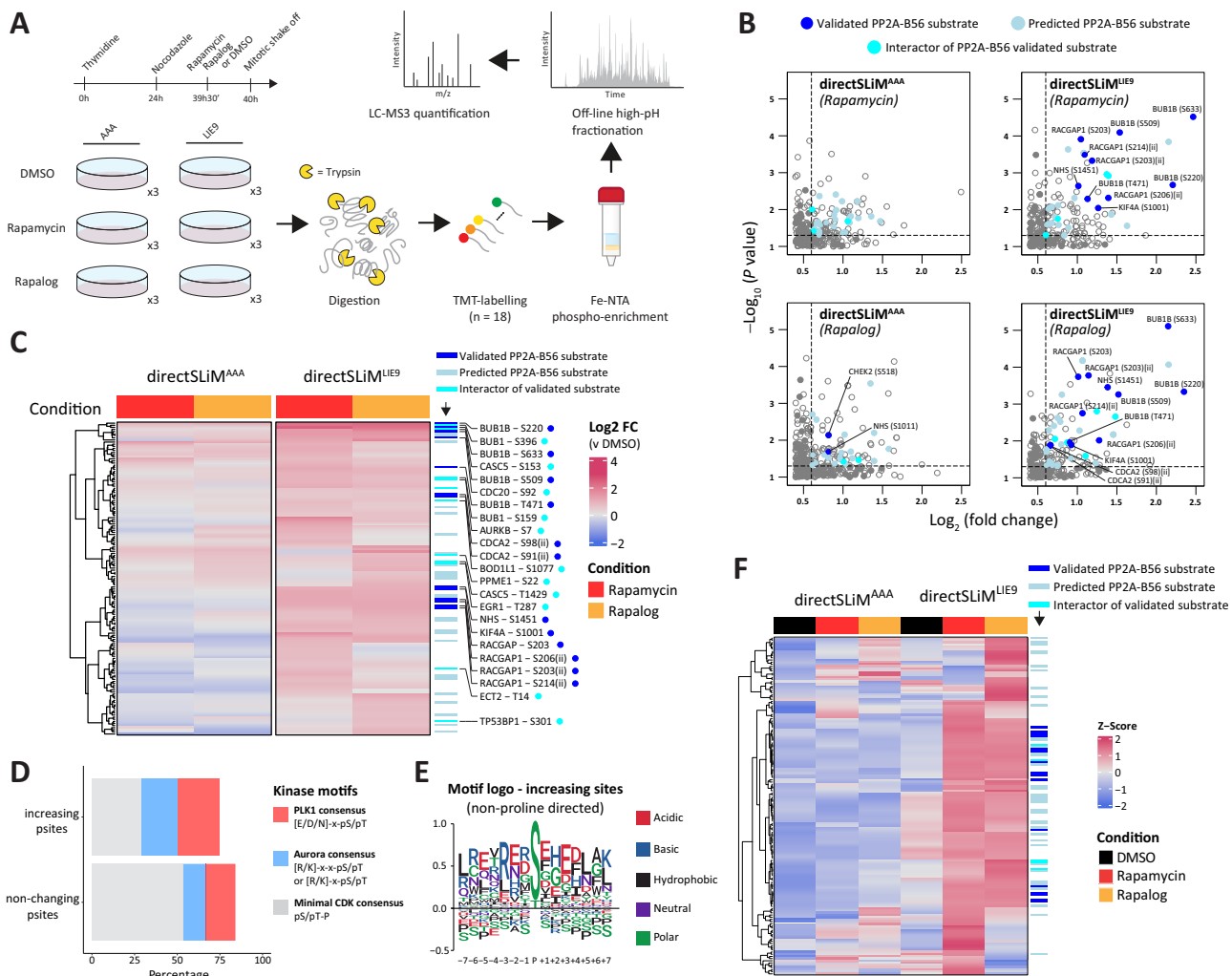

**Fig. 5 | Global protein phosphorylation changes after PP2A-inhibition in directSLiM^LIE9 cells. A** Experimental design: mitotically arrested directSLiM^LIE9 and directSLiM^AAA cells were treated with rapamycin, rapalog or vehicle before harvest ($n = 3$ for each condition). Following cell lysis, proteins were digested, and peptides were TMT-labeled. Next, phosphopeptides were enriched with Fe-NTA spin column, followed by high-pH reversed-phase fractionation and measurement by LC-MS. **B** Zoomed-in right upper quadrant of volcano plot showing upregulated phosphorylation sites after treatment with rapamycin or rapalog. Solid dots are either validated substrates, predicted substrates or interactors of validated PP2A-B56 substrates (see methods for details). Only hits above a -$\text{Log}_{10}(P$ value = 0.05) (Two-sided Student's t-test) and >1.5-fold change are color coded. **C** Heat map of phosphorylation sites upregulated after PP2A-B56 inhibition in directSLiM^LIE9 cells.

Each row represents the intensity of a phosphorylation site ($n = 149$). Intensities are normalised to vehicle treated condition for each cell line to show the relative fold-change upon rapamycin/rapalog addition. Gene names are added for all phospho-sites from validated substrates or interactors of validates substrates. **D** Percentage of Aurora kinase, PLK1 and CDK consensus motifs in the increasing and non-changing phosphorylation sites. **E** Sequence logo of non-proline directed upregulated phosphorylation sites normalized against background (all non-changing phosphosites used as background). **F** Heat map of phosphorylation sites upregulated after PP2A-B56 inhibition in directSLiM^LIE9 cells. Each row represents the z-scored intensity of a phosphorylation site ($n = 155$). The label (ii) refers to phospho-sites in doubly phosphorylated peptides. Source data are provided as a Source Data file.

---

misalignment in metaphase-arrested directSLiM^LIE9 cells treated with rapamycin. This revealed that a significant portion of cells lose their metaphase-aligned chromosomes during the first 30 min of rapamycin treatment, consistent with our fixed analysis (note that chromosome misalignment occurs in all metaphase arrested cells within approximately 2 hours due to cohesion fatigue[43]) (Fig. 6C). Rapamycin treatment in metaphase-arrested directSLiM^LIE1 cells, however, causes a more rapid and often immediate dissociation of chromosome from the metaphase plate (Fig. 6C-D and Supplementary Movies 1-6). This rapamycin-dependent misalignment was also observed in fixed assays and was not caused by loss of centromeric cohesion because it was unaffected by WAPL depletion (Supplementary Fig. 6A).

PP2A-B56 is needed to suppress Aurora B-mediated phosphorylation of the kinetochore-microtubule interface to allow initial attachments to form in prometaphase[13]. In agreement, PP2A-B56

inhibition using directSLiMs increases phosphorylation of NDC80/Hec1, which is a well-established Aurora B substrate that impedes microtubule attachment (Fig. 6E and Supplementary Data 3)[44]. To test if Aurora B activity was responsible for detaching kinetochore-microtubules at metaphase following PP2A-B56 inhibition, we co-treated with the Aurora B inhibitor ZM-447439[45]. ZM-447439 treatment was able to fully rescue kinetochore-microtubule attachments and prevent chromosome misalignment following PP2A-B56 inhibition at metaphase (Fig. 6F-I, Supplementary Fig. 6B–C and Supplementary Movies 7–10). In summary, rapid PP2A-B56 inhibition at metaphase leads to rapid Aurora B-mediated detachment of kinetochore-microtubules. This implies that PP2A-B56 plays a key role in balancing Aurora B activity at metaphase and allowing kinetochores to respond properly to tension exerted by microtubules.

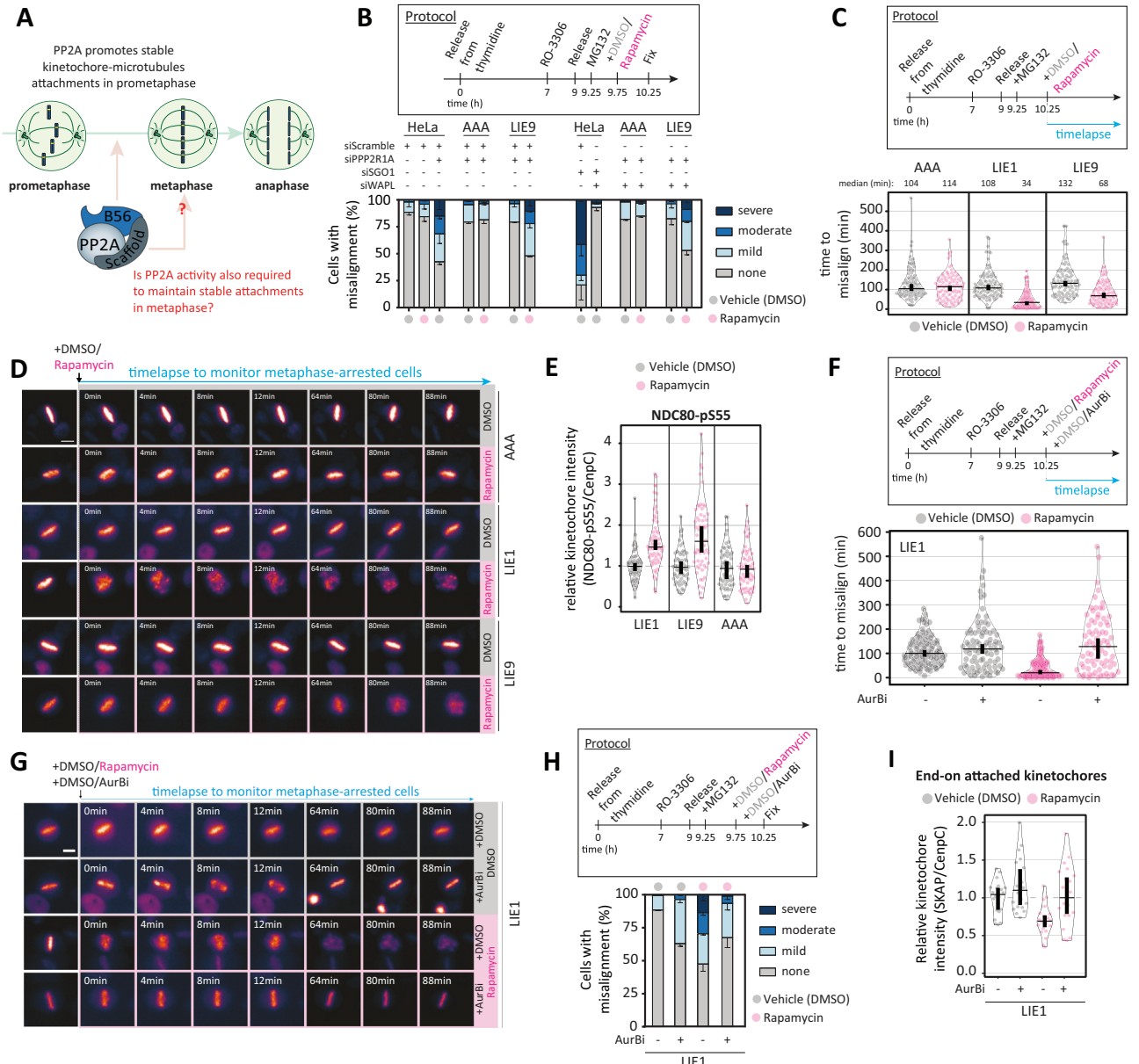

**Fig. 6 | PP2A-B56 is needed to maintain stable kinetochore-microtubules attachment at metaphase. A** Schematic illustrating known and potential contributions of PP2A-B56 in stabilizing kinetochore-microtubule attachments during mitosis. **B** Evaluating the effects on chromosome alignment in HeLa-FRT cells with/without directSLiMs^(LIE9/AAA) expression and treated with vehicle or rapamycin. Top panel: protocol used to visualise chromosome alignment in fixed samples (see Methods for details). Bottom panel: graph showing mean frequencies of chromosome misalignments (±SEM) of 2 experiments, 100 cells quantified per condition per experiment. Individual data points of the bar charts are provided in Source Data file. **C** Evaluating the timings of chromosome misalignments during a metaphase arrest, in HeLa HistH1H4C-TagGFP2 cells expressing the directSLiMs^(LIE1/LIE9/AAA) and treated with vehicle or rapamycin. Top panel shows protocol and bottom panel shows timings of chromosome misalignments from 2 experiments, 50 cells quantified per condition per experiment. **D** Example images of the misalignment timings shown in **C**. **E** Kinetochore intensity of NDC80-pS55 in nocodazole-arrested HeLa-FRT cells with directSLiMs^(LIE1/LIE9/AAA) treated with vehicle or rapamycin. 60 cells quantified from 4

repeats. **F** Evaluating the timings of chromosome misalignments during a metaphase arrest, in HeLa HistH1H4C-TagGFP2 cells expressing the directSLiM^(LIE1) and treated with/without rapamycin and the AurBi ZM-447439, as indicated. Top panel shows protocol and bottom panel shows timings of chromosome misalignments from 2 experiments, 50 cells quantified per condition per experiment. **G** Example images of the misalignment timings shown in **F. H** Fixed chromosome alignment assays. Top panel shows protocol and bottom panel mean mean frequencies of chromosome misalignments (±SEM) of 2 experiments, 100 cells quantified per condition per experiment. Individual data points of the bar charts are provided in Source Data file. **I** Kinetochore intensities of SKAP from cells treated as in H. SKAP is a marker of mature end-kinetochore microtubules attachments[22]. 20 cells quantified from 2 repeats. Source data are provided as a Source Data file. Data information: Violin plots show the distributions of misalignment timings between cells. For each violin plot, each dot represents an individual cell, the horizontal line represents the median and the vertical one the 95% CI of the median, which can be used for statistical comparison of different conditions (see Methods). Scale bar: 10 µm.

## Discussion

Here we present a chemical-genetic system that can rapidly block the substrate binding pocket on PP2A-B56, thereby effectively inhibiting phosphatase activity. This is a powerful approach that can specifically inhibit PP2A-B56 function within seconds of drug addition. We use this

to identify PP2A-B56 substrates during mitosis and to uncover a role of PP2A-B56 in stabilising kinetochore-microtubule attachments at metaphase. It should now be possible to use this directSLiMs system to carefully study PP2A-B56 function during any other cell cycle phase or biological process.

The full list of phospho-sites that exhibit a greater than 1.5-fold change in at least one of our datasets is displayed in Supplementary Data 3. This list contains 513 increasing phospho-sites, across a total of 392 unique proteins. A previous mitotic PP2A-B56 substrate screen by Kruse et al. identified 33 of these increasing phospho-sites (6.4%) and other sites on 75 of these proteins (19.1%)[25]. Therefore, while there is overlap between these studies, as expected, our dataset greatly expands the list of putative PP2A-B56 substrates during mitosis. It also focusses on substrates that change relatively quickly following PP2A-B56 inhibition (30 min inhibition compared to 12 h in Kruse et al.), which are more likely to be direct substrates.

In agreement with the Kruse et al. study[25], we see that the PP2A-B56 regulated sites are not enriched for pThr over pSer, however, proline directed sites ([pS/pT]-P) are strongly disfavoured (Supplementary Data 3). This could ensure that active PP2A-B56 can function during mitosis without extinguishing CDK-mediated phosphorylations. However, we do still detect many potential CDK-sites increasing after PP2A-B56 inhibition, including the well-characterised substrates BUBR1 (pT620) and BUB1 (pT609). Considering that PP2A-B56 binding strength and position can both influence dephosphorylation efficiency[25], we speculate that LxxIxE motifs could evolve to allow specific CDK sites to be regulated prior to anaphase when the majority are dephosphorylated. This might explain why the close positioning between the PP2A-B56 binding site (BUBR1 aa669-680) and the CDK1-mediated PLK1 binding site (BUBR1-pT620) has been so well conserved throughout evolution[17].

An interesting observation from our proteomic datasets is that most of the phospho-sites that increase following PP2A-B56 inhibition do so by a relatively low fold-change: only a few sites, mainly on the BUB complex, exhibit a greater that 3-fold increase (Figs. 2B and 5B). This is unlikely to reflect ineffective PP2A-B56 inhibition because direct comparison to local PP2A inhibition, via mutation of the kinetochore receptor BUBR1, revealed that most kinetochore targets increased similarly (Supplementary Fig. 7). This implies that PP2A-B56 does not generally keep sites fully dephosphorylated, but rather it supresses sites that are already phosphorylated, in keeping with the fact that it is recruited to phosphorylated substrates[7]. We hypothesise that this reflects a role for PP2A-B56 in the continuous turnover of phosphorylation sites, causing them to dynamically switch between the phosphorylated and dephosphorylated state. In this scenario, inhibiting PP2A-B56 may only cause a small increase in substrate phosphorylation, even though the underlying dynamics on each substrate molecule may change dramatically (e.g. from continually switching "on" and "off" to statically just "on"). There are many important signalling properties that rely on rapid phosphorylation/dephosphorylation dynamics, as discussed here[46], including signals that must respond rapidly to changing kinase activity. We speculate that a key function of PP2A-B56 at kinetochores is to allow kinetochore signals to change state quickly, for example, in response to microtubule attachment and/or tension, when Aurora B and/or MPS1 are inhibited[13].

We use the directSLiMs system to demonstrate that PP2A-B56 is needed to counteract Aurora B activity to maintain kinetochore-microtubule attachments at metaphase, in addition to its previous known role in stabilising initial kinetochore-microtubule attachments in prometaphase[9,10,40]. This is important because it implies that PP2A-B56 is required for proper tension-sensing at kinetochores: a process that is still poorly understood[47]. Aurora B activity is reduced at kinetochores that come under tension, thereby stabilising kinetochore-microtubule attachments on chromosomes that have aligned to the metaphase plate. We propose that PP2A-B56 is needed to allow kinetochores to respond correctly to the reductions in Aurora B activity induced by tension. This could be either direct or indirect. For example, PP1 recruitment to KNL1 is inhibited by Aurora B[48], and this has been proposed to contribute to tension sensing[49]. PP2A-B56 can antagonise Aurora B to enhance PP1 kinetochore recruitment[15],

therefore this could at least partially explain how PP2A-B56 controls kinetochore-microtubule stability at metaphase. It is also important to note that the effects of PP2A-B56 could reflect inhibition of Aurora B itself and/or its downstream substrates. Our sequence analysis of the phosphorylated peptides that increase following rapamycin/rapalog treatment suggests that PP2A-B56 generally antagonises the mitotic kinases Aurora A/B, PLK1 and/or MPS1, as suggested by numerous previous studies[9,10,15,17-21,34,40]. Whether these phosphorylation sites increase because of increased kinase activity, decreased substrate dephosphorylation, or a combination of both, remains to be determined.

The directSLiMs approach is also easily adaptable to inhibit other SLiM-binding interfaces on any other proteins. Care just needs to be taken to identify SLiMs with the optimal binding strength, so that inhibition is only observed upon drug addition. However, there may still be situations when strong binding SLiMs are preferred, for example, if strong inhibition or very rapid inhibition kinetics are required. A good example is the use of directSLiM[LIE1] cells to cause immediate detachment of chromosomes at metaphase, which is not observed in directSLiM[LIE9] cells (Fig. 6C–D). Care also needs to be taken to assess the effects of drug alone, especially if rapamycin is used. In mitosis we observed very few phosphorylation changes within 30 minutes rapamycin treatment (Fig. 2B–C), but this may be due to the fact that TORC1 activity is strongly inhibited during a mitotic arrest[32,33]. Rapalog does not inhibit TORC1 and this also works well, albeit with slightly slower kinetics (Fig. 1G–H). Other chemically-induced dimerisation systems could also be used, if required[50].

In summary, here we present a chemical-genetic system to inhibit and characterise PP2A-B56, revealing a role for this phosphatase in maintaining stable kinetochore-microtubule attachments at metaphase. This system should be expandable to inhibit other phosphatase complexes that also rely on SLiM-based substrate recognition[3]. The directSLiMs approach may therefore ultimately prove as valuable for characterising phosphatases as the "bump-and-hole" approach has proved for characterising kinases that lack small molecules inhibitors[51]. In fact, it could be more widely applied to characterise many other enzymes or proteins that are regulated by SLiM-based interactions. The eukaryotic linear motifs resource currently lists hundreds of annotated SLiM classes, regulating a wide range of processes including protein transcription and translation, cellular trafficking, cell cycle, and protein degradation[26]. DirectSLiMs could therefore prove to be a powerful approach to dissect the role of these different SLiM classes once the SLiM-binding domains have been identified and validated.

## Methods

### Cell culture and reagents

All cell lines used in this study were derived from HeLa-FRT cells (also known as HeLa Flp-in: a gift from S Taylor, University of Manchester, UK)[52] apart from Phoenix Ampho cells (ATCC, CRL-3213). Cells were authenticated by STR profiling (Eurofins). Cells were cultured in full growth media – DMEM supplemented with 9% FBS and 50 μg/ml penicillin/streptomycin. Every 4–8 weeks, cells were screened to ensure a mycoplasma free culture. The following drugs were used (at the indicated concentrations throughout): Doxycycline (1 μg/ml), thymidine (2 mM), nocodazole (3.3 μM) and MG132 (10 μM) were purchased from Sigma Aldrich; rapamycin (1 μM) from LC Labs; A/C heterodimeriser (or Rapalog, 500 nM) from Clontech; puromycin (1 μg/ml) and hygromycin B (200 μg/ml) from Santa Cruz Biotechnology; RO-3306 (10 μM) from Tocris; the SiR-DNA far-red DNA probe (1:10,000) from Spirochrome; the AuroraB kinase inhibitor ZM-447439 (2 μM) from Cayman Chemicals.

### Plasmids and cloning

Plasmids identified with a DUXXXXX number were cloned by MRC PPU Reagents & Services Laboratory and can be ordered from MRC PPU Reagents (https://mrcppureagents.dundee.ac.uk/)

To clone the directSLiMs construct pcDNA5-LIE1-VSV-FRB-T2A-FKBP-3xFLAG-PPP2R1A (named LIE1[FRB]-T2A-[FKBP]R1A hereafter, with LIE1 sequence STLPTIHEEEEE), synthesised fragments for siRNA-resistant PPP2R1A and for LIE1-VSV-FRB-T2A-FKBP (Genestrings, Thermofisher) were subcloned into pcDNA5/FRT/TO using KpnI-BamHI and BamHI-ApaI respectively. To generate AAA[FRB]-T2A-[FKBP]R1A, the LPTI sequence in the LIE1 feature of LIE1[FRB]-T2A-[FKBP]R1A was replaced with AAAA by site-directed mutagenesis (STAAAAHEEEEE - DU71843, MRC PPU Reagents & Services Laboratory - https://mrcppureagents.dundee.ac.uk/). To generate LIE2-7[FRB]-T2A-[FKBP]R1A constructs, the LIE1 feature in LIE1[FRB]-T2A-[FKBP]R1A was replaced with the sequence for LIE2 (QELYSIPEDQEPED, DU75900), LIE3 (QELPPVQEDDE, DU75903), LIE4 (GYLPTLNEDEE, DU75898), LIE5 (KKLEPIIEDE, DU75904), LIE6 (SGCSPIEEEAH, DU75901) or LIE7 (SGASPIEEEAH, DU75902) by annealing the appropriate oligo pair and ligating into LIE1[FRB]-T2A-[FKBP]R1A digested with KpnI and AvrII (MRC PPU Reagents & Services Laboratory - https://mrcppureagents.dundee.ac.uk/). To generate LIE8[FRB]-T2A-[FKBP]R1A, the LIE1 feature of LIE1[FRB]-T2A-[FKBP]R1A was replaced with STLPTIHEAAAA by site-directed mutagenesis (DU75852, MRC PPU Reagents & Services Laboratory - https://mrcppureagents.dundee.ac.uk/). Site-directed mutagenesis with specific primers (Sigma-Alrich) was performed on LIE8[FRB]-T2A-[FKBP]R1A to generate: LIE9[FRB]-T2A-[FKBP]R1A (STLPTIHEEAAA; forward: 5′- GCCCACAATTCATGAAGAAGCAGCGGCAGGAGG-3′, reverse: 5′- CCTCCTGCCGCTGCTTCTTCATGAATTGTGGGC-3′), LIE10[FRB]-T2A-[FKBP]R1A (STLPTIHEEEAA; forward; 5′- GCTGCCCACAATTCATGAAGAAGAGGCGGCAGGAGGAGG-3′, reverse: 5′- CCTCCTCCTGCCGCCTCTTCTTCATGAATTGTGGGCAGC-3′), LIE11[FRB]-T2A-[FKBP]R1A (STLPTIHEEEEA; forward: 5′- GAAGAAGAGGAAGCAGGAGGAGGTTCCGG-3′, reverse: 5′- CCGGAACCTCCTCCTGCTTCCTCTTCTTC-3′), LIE12[FRB]-T2A-[FKBP]R1A (STLPTIHEAEAA; forward: 5′- CAATTCATGAAGCTGAGGCGGCAGGAGGAGGTTCC-3′, reverse: 5′- GGAACCTCCTCCTGCCGCCTCAGCTTCATGAATTG-3′), LIE13[FRB]-T2A-[FKBP]R1A (STLPTIHEAEAA; forward: 5′- CATGAAGCTGCAGAGGCAGGAGGAGGTTCC-3′, reverse: 5′- GGAACCTCCTCCTGCCTCTGCAGCTTCATG-3′) and LIE14[FRB]-T2A-[FKBP]R1A (STLPTIHEAAAE; forward: 5′- GAAGCTGCAGCGGAAGGAGGAGGTT CCGG-3′, reverse: 5′- CCGGAACCTCCTCCTTCCGCTGCAGCTTC-3′). All final plasmids were verified by sequencing carried out by MRC PPU Reagents and Services Sequencing Laboratory (https://dnaseq.co.uk/). To generate doxycycline-inducible PPP2R1A shRNAs we used the pSuperior system (OligoEngine) to create a pSuperior-shPPP2R1A-33 plasmid (pSP-shR1A-33). Oligos containing the shRNA sequence (forward: 5′- GATCCCCTTTTCCACTAGCTTCTTCATTCAAGAGATGAAGAAGCTAGT GGAAAATTTTTA and reverse: 3′-AGCTTAAAAATTTTCCACTAGCTTCT TCATCTCTTGAATGAAGAAGCTAGTGGAAAAGGG) were annealed and ligated into pSuperior Retro Puro (OligoEngine) using HindIII and BglII. The CRISPaint constructs pCas9-mCherry-Frame1, pCas9-mCherry-HistH1H4C, and pCRISPaint-TagGFP2-PuroR were purchased from Addgene (Kit #1000000086)[53]. pcDNA5-YFP-BUBR1[WT] expressing a N-terminally YFP-tagged and siRNA-resistant wild-type BUBR1 was described previously[15]. pcDNA5-YFP-BUBR1[ΔPP2A(ΔK)] (also called BUBR1[ΔKARD]), lacking amino acids 664–681 of BUBR1, was described previously[15]. Cloning of pMESV$_\Psi$-mCherry-B56γ$_1$ has been described previously[16]. A similar strategy was also used to clone pMESV$_\Psi$-mCherry-B56α.

## Gene expression
HeLa-FRT cells were stably generated to allow doxycycline-inducible expression of all the directSLiMs constructs. Cells were transfected with the relevant pcDNA5/FRT/TO directSLiMs plasmid and the Flp recombinase pOG44 (Thermo Fisher) using Fugene HD (Promega) according to the manufacturer's instructions. Subsequently, stable integrants at the FRT locus were selected using hygromycin B for at least 2 weeks. Cells expressing mCherry-B56α or mCherry-B56γ were generated by viral integration of the pMESV$_\Psi$ constructs into the genome of HeLa-FRT cells, followed by puromycin selection. These cells were then used to stably express the LIE1[FRB]-T2A-[FKBP]R1A construct by following the same procedure described above.

## Generation of HeLa pSP-PPP2R1A cell lines
Phoenix Ampho 293 cells (ATCC) were transfected with pSP-shR1A-33 using Fugene HD (Promega) according to the manufacturer's instructions. After 24 h, cells were washed and placed in fresh full growth media. Forty-eight hours after transfection, virus containing media was harvested and filtered using a 0.45uM filter before adding to HeLa-FRT cells. Cells were infected with virus 3 times before the media was replaced with full growth media for 48 h. Cells were then selected with puromycin for 3 days. HeLa pSP-PPP2R1A cells were then used to stably express doxycycline-inducible directSLiMs constructs by following the same procedure described above.

## Generation of HistH1H4C-tagged cell lines
The endogenous HistH1H4C in HeLa-FRT cells was tagged with TagGFP2 by using the CRISPaint method[53]. Briefly, cells were transiently transfected with pCas9-mCherry-Frame1 frame selector, pCas9-mCherry-HistH1H4C target selector and pCRISPaint-TagGFP2-PuroR donor plasmid at a ratio of respectively 1:1:2, using Fugene HD (Promega) according to the manufacturer's instructions. After 2 days from transfection, cells were selected using puromycin for 2 weeks. These cells were then used to stably express doxycycline-inducible directSLiMs[LIE1/LIE9/AAA] constructs by following the same procedure described above.

## Gene knockdown
For all experiments involving expression of the directSLiMs constructs in HeLa-FRT or HeLa pSP-PPP2R1A cells, the endogenous mRNA of PPP2R1A was knocked down (siPPP2R1A: 5′-CCACCAAGCA-CAUGCUACC-3′, dTdT overhang) and replaced with an siRNA-resistant FKBP-FLAG-tagged mutant expressed from the doxycycline-inducible directSLiMs construct. The other siRNAs used in this study were: siBUBR1 (5′-AGAUCCUGGCUAACUGUUC-3, UU overhang′), siSGO1 (5′-GAUGACAGCUCCAGAAAUU-3′, UU overhang), siWAPL (5′- GAGA-GAUGUUUACGAGUUU-3′, UU overhang) and siScramble (control siRNA: 5′- AAGCGCGCTTTGTAGGATTCG-3′, UU overhang). All synthesised siRNAs (Sigma-Aldrich) were used at 20 nM final concentration. All siRNAs were transfected using Lipofectamine® RNAiMAX Transfection Reagent (Thermo Fisher) according to the manufacturer's instructions. After 16 h of knockdown, cells were arrested with thymidine for 24 h. Doxycycline was used to induce the expression of the directSLiMs constructs—and to express PPP2R1A shRNA construct in HeLa pSP-PPP2R1A cells—during and following the thymidine block. Cells were then released from thymidine block into full growth media supplemented with doxycycline and, when appropriate, nocodazole for 5-7 hours for live imaging or 8.5 hours before processing for fixed analysis. Data was obtained using the following cell lines expressing directSLiMs: HeLa-FRT (Fig. 1 and Supplementary Fig. 1B-C); HeLa HistH1H4C-TagGFP2 cells (Fig. 6C–D and F–G and Supplementary Fig. 6B); HeLa-FRT mCherry-B56a or mCherry B56g (Supplementary Fig. 1A); HeLa-FRT YFP-BubR1 WT or dK (Supplementary Fig. 7); HeLa-FRT pSP-PPP2R1A cells (all other figures).

## Immunofluorescence
Cells plated on High Precision 1.5H 12-mm coverslips (Marienfeld) were fixed with 4% paraformaldehyde (PFA) in PBS for 10 min or pre-extracted with 0.1% Triton X-100 in PEM (100 mM PIPES, pH 6.8, 1 mM MgCl$_2$ and 5 mM EGTA) for 1 minute before addition of 4% PFA for 10 minutes. Pre-extraction was only performed in cells shown in Fig. 1E-H and Supplementary Fig. 1B-C. After fixation, coverslips were washed with PBS and blocked with 3% BSA in PBS + 0.5% Triton X-100 for 30 min, incubated with primary antibodies overnight at 4 °C, washed with PBS and incubated with secondary antibodies plus DAPI (4,6-

diamidino2-phenylindole, Thermo Fisher) for an additional 2–4 hours at room temperature in the dark. Coverslips were washed with PBS and mounted on glass slides using ProLong antifade reagent (Molecular Probes). All images were acquired on a DeltaVision Core or Elite system equipped with a heated 37 °C chamber, using a CoolSNAP HQ or HQ2 camera (Photometrics) with a 100x/1.40 NA U Plan S Apochromat objective using softWoRx software (Applied precision), or on a Nikon Ti2-E Eclipse system equipped with a heated 37 °C Okolab chamber, using a Kinetix camera (Teledyne Photometrics) with a CFI Plan Apochromat λD 100x/1.45 NA oil objective (Nikon) with NIS-Elements AR software (Nikon). Images were acquired at 1×1 binning and processed using softWorx software, Nikon NIS Elements and ImageJ (National Institutes of Health). Mitotic cells arrested in early prometaphase were selected for imaging based on good expression of FLAG in the cytoplasm. All immunofluorescence images displayed are maximum intensity projections of deconvolved stacks and were chosen to closely represent the median quantified data. Figure panels were creating using Omero (http://openmicroscopy.org).

The following primary antibodies (all diluted in 3% BSA in PBS) were used at the final concentration indicated: guinea pig anti-CENP-C (PD030 from Caltag + Medsystems, 1:5000), rabbit anti-BUB1 (A300-373A from Bethyl, 1:1000), mouse anti-BUBR1(8G1) (05-898 from Millipore, 1:1000), rabbit anti-BUBR1 (A300-386A from Bethyl, 1:1000), rabbit anti-KNL1 (ab70537 from abcam, 1:1000), rabbit anti-SKAP(KNSTRN) (HPA042027 from Atlas Antibodies, 1:1000), rabbit anti-HEC1-pSer55 (GTX70017 from Genetex, 1:500), mouse anti-FLAG(M2) (F3165-.2MG from Sigma, 1:1000).

The rabbit anti-pMELT-KNL1 antibody is directed against phospho-Thr 943 and -Thr 1155 of human KNL1[15] (1:1000 – gift from G. Kops, Hubrecht, NL). The rabbit anti-pRVSF-KNL1 was raised against phospho-Ser 60 of human KNL1 (using the peptide C-CKKNSRRV[pS]FADTIK, custom raised by Biomatik, 1:500). The rabbit anti-BUBR1-pT620 antibody was raised against phospho-Thr 620 of human BUBR1 using the peptide C-AARFVS[pT]PFHE (custom raised by Moravian, 1:1000)[17]. The rabbit anti-BUB1-pT609 antibody was raised against phospho-Thr 609 of human BUB1 using the peptide C-AQLAS[pT]PFHKLPVES (custom raised by Biomatik, 1:2000)[17]. The rabbit anti-BUB1-pT461 against phospho-Thr 461 of human BUB1 was a gift from M. Bollen (Leuven, BE; 1:500)[37].

Secondary antibodies used were highly-cross absorbed goat anti-chicken Alexa Fluor 488 (A-11039), goat anti-rabbit Alexa Fluor 568 (A-11036), goat anti-mouse Alexa Fluor 488 (A-11029), goat anti-mouse Alexa Fluor 568 (A-11031), goat anti-guinea pig Alexa Fluor 647 (A-21450), donkey anti-rabbit Alexa Fluor 647 (A-31573) or donkey anti-mouse Alexa Fluor 647 (A-31571) all used at 1:1000 (Thermo Fisher).

## Immunoprecipitation and Western blotting

Endogenous PPP2R1A was knocked down with siRNA as above. Doxycycline was added 1 h after siRNA transfection. After 24 h, cells were treated with thymidine and doxycycline for 24 h and then released into fresh media supplemented with nocodazole and doxycycline for 17 h. After addition of DMSO or Rapamycin for 30 min, mitotic cells were isolated by mitotic shake off and washed with ice-cold PBS. For α-FLAG-immunoprecipitations, cells were lysed in lysis buffer [50 mM Tris, pH 7.5, 150 mM NaCl, 1 mM EDTA, 1% TX-100, phosStop and complete protease inhibitor cocktail, EDTA-free (both Roche)] on ice and then incubated at 4 °C on a rotating wheel for 20 min before centrifugation to remove insoluble material. 2-3 mg lysate was incubated with 20µl FLAG-M2 magnetic beads (M8823, Sigma) for 3 hr at 4 °C on a rotating wheel. The beads were washed 3x with wash buffer [50 mM Tris, pH 7.5, 150 mM NaCl, phosStop and complete protease inhibitor cocktail containing EDTA (both Roche)]. The sample was eluted by boiling the beads in SDS-PAGE gel loading buffer (62.5 mM Tris, 2.5% SDS, 10% glycerol and 5% 2-mercaptoethanol) for 5 min. Samples were processed for SDS-PAGE and immunoblotted using

standard protocols. For mCherry precipitations, the same procedure was used except: cells were lysed in 50 mM Tris, pH 7.5, 150 mM NaCl, 1 mM EDTA, 1% NP40, 5% glycerol, phosStop and complete protease inhibitor cocktail, EDTA-free; the lysate was incubated with 25ul mCherry-sepharose beads (MRC PPU); immunoprecipitates were washed with the same buffer used for cell lysis.

The following primary antibodies (all diluted in 5% non-fat milk in TBST) were used at the final concentration indicated: mouse anti-B56α (BD Biosciences 610615, 1:2000), mouse anti-B56γ (Santa Cruz Biotechnology sc-374379, 1:1000), mouse anti-B56δ (Santa Cruz Biotechnology sc-271363, 1:1000), rabbit B56ε (Aviva ARP56694-P050, 1:1000), mouse anti-PPP2CA (EMD Millipore 05-421, 1:1000), rabbit anti-PPP2R1A (Genetex GTX102206, 1:1000), rabbit anti-GEF-H1 (Abcam 155785, 1:1000), rabbit anti-BubR1 (Bethyl Laboratories A300-386A, 1:1000), rabbit anti-CDCA2 (Repoman, Sigma HPA030049, 1:1000), mouse anti-FLAG (Sigma F1804, 1:5000), rabbit anti-mCherry (Genetex GTX128508, 1:1000), rabbit anti-Actin (Sigma A2066, 1:2500) and mouse anti-Tubulin (Sigma, T5168, 1:10000). The secondary antibodies were goat anti-mouse IgG HRP conjugate (Bio-Rad 170–6516, 1:2000), goat anti-rabbit IgG HRP conjugate (BioRad 170–6515, 1:5000), IRDye® 800CW goat anti-mouse (LICORbio, 926-32210, 1:10000) and IRDye 800CW donkey anti-rabbit (LICORbio, 926-32213, 1:10000). The secondary antibody used to detect B56α, B56γ, B56δ and FLAG was goat anti-mouse light chain specific HRP-conjugated (Sigma, AP200P, 1:1000).

## Chromosome alignment assays

To observe live chromosome alignment and determine mitotic cell fates and timing, cells were plated in 8-well or 18-well chamber slides (ibidi), released from thymidine block for 5–6 hours, incubated with SiR-DNA far-red DNA probe (1:10000, Spirochrome; to prevent toxicity[54]) in full growth media for 15 min, and treated with DMSO or rapamycin prior to imaging. Images were captured, after treatment with DMSO or rapamycin, every 4 minutes for 16 hours with a CFI Plan Apochromat λD 40x/0.95 NA air objective (Nikon) using a Nikon Ti2-E Eclipse with a Kinetix camera (Teledyne Photometrics) at 4 × 4 binning, 10 z-stacks with a step size of 1.50 μm. Selected cells were scored based on the following mitotic events: cohesion fatigue, cell division or cell death following chromosome alignment or not.

To observe chromosome alignment in fixed-cell experiments, cells were released from thymidine block for 7 hours before being treated for 2 hours with RO-3306 to synchronise cells at the G2/M boundary. Cells were then washed three times and incubated for 15 minutes with full growth media before addition of MG132 to prevent mitotic exit. After 30′ from MG132 addition, cells were treated with DMSO or rapamycin with or without ZM-447439 and fixed 30′ after the treatment. Fixed cells were stained as described above and imaged on a Zeiss Axio Observer with a CMOS Orca flash 4.0 camera at 4 × 4 binning, using a Plan-apochromat 20×/0.4 air objective, or on a Nikon Ti2-E Eclipse with Kinetix camera (Teledyne Photometrics) at 4 × 4 binning, using a CFI Plan Apochromat λD 20x/0.8 NA air objective. Cells with good expression of FLAG-tagged PPP2R1A were scored based on the number of misaligned chromosomes as aligned (0 misaligned chromosomes, with a visible metaphase plate), mild (1–2), moderate (3–5) or severe ( > 6).

To determine chromosome misalignment timings during an arrest in metaphase, HeLa HistH1H4C-TagGFP2 cells were plated in 8-well slides (ibidi), released from thymidine block for 7 hours and treated with RO-3306 and MG132 as described above, to enrich metaphase-arrested cells. After 1 h from MG132 treatment, cells were treated with DMSO or rapamycin with or without ZM-447439. To monitor chromosome misalignments of those cells arrested in metaphase prior to DMSO/rapamycin treatment, the imaging started 30′ after MG132 treatment, paused during the DMSO or rapamycin treatment and then resumed. Images were captured every 4 minutes for

12 hours with a CFI Plan Apochromat λD 40x/0.95 NA air objective (Nikon) using a Nikon Ti2-E Eclipse with a Kinetix camera (Teledyne Photometrics) at 4 × 4 binning, 10 z-stacks with a step size of 1.50 μm.

### Image quantification

For quantification of kinetochore protein levels, images of similarly stained experiments were acquired with identical illumination settings and analysed using an ImageJ macro, as described previously[55]. Briefly, the macro performs a threshold and selection of all the kinetochores, using the DAPI and CenpC channels. To generate kinetochore masks, the macro applied a convolution filter to the CenpC channel and perform a threshold selection. The resulting masks were then increased by 1 pixel (to ensure complete kinetochore selection). These masks were then used to calculate the relative mean kinetochore intensity of FLAG-PPP2R1A, Bub1, BubR1, Knl1, Bub1-pT609, Bub1-pT461, BubR1-pT620, Knl1-pMELT, Knl1-pRVSF, Hec1-pS55, SKAP and CenpC. Fluorescence intensities at kinetochores were normalised to CenpC (i.e. kinetochore marker)[56].

### Phosphoproteomics: Cell culture and sample preparation

After 24 h of knockdown and doxycycline addition, cells were arrested with thymidine for 24 h. Cells were then released from thymidine block into full growth media with doxycycline and nocodazole for 15 hours. Cells were treated with either DMSO (0.1%), Rapamycin or rapalog for 30 minutes before harvesting (three biological replicates for each condition). Mitotic cells were collected in their own media by shake-off, followed by a quick wash in plain DMEM containing doxycycline and the respective drug. Cells were then lysed with 2% SDS in PBS supplemented with protease and phosphatase inhibitors and snap-frozen in liquid nitrogen.

Cell lysates were subsequently cleared by sonication and nucleic acids were digested with Benzonase for 30 min at 37 °C. Proteins were precipitated with acetone and digested into peptides with trypsin (1:100 enzyme to protein ratio) for 16 h at 37 °C, followed by a second round of digestion for 4 h at 37 °C. Next, peptides were acidified with formic acid (final concentration of 3%) and desalted with C18 Quik-Prep® Micro SpinColumns™ from Harvard Apparatus. Briefly, columns were conditioned with 100% acetonitrile and then equilibrated with 0.5% formic acid. Peptides were loaded into the columns and washed twice with 0.5% formic acid followed by peptide elution using 80% acetonitrile in 0.5% formic acid. Peptides were dried in a vacuum concentrator at 30 °C. Next, each sample was dissolved in 100 mM TEAB, mixed with 0.25 mg of TMTpro label and incubated for 1 h at room temperature. Labelling reaction was quenched by adding 2.5 uL of 5% hydroxylamine to each sample and incubating for 15 minutes at 37 °C. Next, all samples were pooled together and dried before C18 desalting using 50 mg Sep-Pak C18 columns. Briefly, the TMT-labeled peptides were loaded into the columns, followed by washes with 0.5% formic acid and elution with 80% acetonitrile in 0.5% acetic acid.

Depending on the experiment, two different methods for phosphopeptide enrichment were used. For the experiment in Fig. 2, peptides were mixed with MagResyn Ti-IMAC HP in 80% acetonitrile, 5% trifluoroacetic acid (TFA), 5% glycolic acid and incubated for 20 min at 25 °C. Beads were washed with 80% acetonitrile, 1% TFA for 2 min, followed by a second wash with 10% acetonitrile, 0.2% TFA. Phosphopeptides were eluted twice with 1% ammonium hydroxide for 15 min, followed by a final elution with 50% acetonitrile, 1% ammonium hydroxide for 1 h. To increase phosho-peptide recovery, the flow-through was mixed with a new batch of MagResyn Ti-IMAC HP beads followed by the same steps previously described. For the experiments in Fig. 5, phospho-peptides were enriched using the High-Select Fe-NTA Phosphopeptide Enrichment Kit using the manufacturer instructions. Briefly, peptides were reconstituted in Binding/Wash Buffer and loaded into the Fe-NTA columns with an incubation time of 30 min at room temperature, gently mixing the resin every 10 min. Next, the

column was washed a total of three times with Binding/Wash Buffer, before a final wash with LC-MS grade water. Phosphopeptides were eluted with Elution Buffer and acidified to 5% formic acid.

To ensure removal of magnetic beads or resin, the phosphopeptides were dried and desalted using C18 QuikPrep® Micro SpinColumns™ from Harvard Apparatus as previously described. To achieve a deep coverage of the phosphoproteome, peptides were fractionated using high-pH reversed-phase chromatography. Briefly, peptides were injected into a 1.0 ×100 mm column packed with 1.7 μm BEH particles with a 130 Å pore size coated with C18. For elution, we used a 15–80% B gradient using the following mobile phases: A, was 10 mM ammonium formate pH 9.0 in water and B was 10% ammonium formate pH 90 and 90% acetonitrile. Peptides were eluted, dried and stored at -20 °C until measurement by LC-MS.

### Phosphoproteomics: Data acquisition

The experiment in Fig. 2 was measured using a Dionex Ultimate 3000 HPLC coupled to an Orbitrap Fusion Tribrid mass spectrometer. Peptides from 14 fractions were loaded and separated using 75 μm × 50 cm EASY-Spray column with 2 μm sized particles. The column was kept at 50 °C using an EASY-Spray source. The following mobile phases were used for the gradient: Buffer A consisted of 0.1% formic acid in LC-MS grade water and buffer B consisted of 80% acetonitrile and 0.1% formic acid. With a flow rate of 300 μL/min a gradient was applied, starting from 5% to 35% B in 130 minutes, followed by a 20 min wash with 98% B and a subsequent column equilibration with 5% for 20 min. A voltage of 2.2 kV was set for electrospray ionization and a MS1 scan on the Orbitrap was acquired at a 120,000 resolution with a maximum injection time of 50 ms with a scan range of 380–1500 m/z. The cycle time (time between MS1 scans) was set to 3 s. For peptide identification, precursors ions with a charge state of 2–6 were isolated (0.7 m/z isolation window) for CID fragmentation at 35% and subsequent MS2 on the Orbitrap at a 30,000 resolution with a maximum injection time of 60 ms. Dynamic exclusion was set for a duration of 45 s. 10 precursor fragments were selected for synchronous precursor selection (SPS) using an isolation window of 0.7 m/z and subjected to HCD at 65% for release of the TMTpro reporter tag and measured in the Orbitrap at 60,000 resolution with a maximum injection time of 105 ms.

The experiments in Fig. 5 were measured using a Vanquish Neo UHPLC coupled to an Orbitrap Eclipse Tribrid mass spectrometer. Peptides from 16 fractions were separated using a 75 μm × 50 cm EASY-Spray PepMap Neo column with 2 μm sized particles coated with C18. The column was kept at 50 °C using an EASY-Spray source. The following mobile phases were used for the gradient: Buffer A consisted of 0.1% formic acid in LC-MS grade water and buffer B consisted of 80% acetonitrile and 0.1% formic acid. A flow rate of 300 μL/min was used to apply a gradient from 2% to 40% B in 150 min, followed by a wash with 95% B for 20 minutes. A voltage of 1.9 kV was set for electrospray ionization and an MS1 scan on the Orbitrap was acquired at a 120,000 resolution with a maximum injection time of 50 ms and a scan range of 380–1500 m/z. The cycle time (time between MS1 scans) was set to 3 s. For peptide identification, precursors with charge state of 2–7 were isolated (0.7 m/z isolation window) for HCD fragmentation at 28% and subsequent MS2 on the ion trap with a maximum injection time of 50 ms. Dynamic exclusion was set for a duration of 70 s. 5 precursor fragments were selected for SPS using an isolation window of 0.7 m/z and subjected to HCD at 55% for release of the TMTpro reporter tag, followed by measurement in the Orbitrap at 50,000 resolution with a maximum injection time of 90 ms.

### Phosphoproteomics: Data analysis

Raw data were processed using MaxQuant (version 2.4.9.0) with the default settings for a TMTpro-18plex experiment. A maximum of 2 missed cleavages were allowed and the enzyme parameter was set to 'Trypsin/P'. Carbamidomethyl (C) was added as a fixed modification

while Oxidation (M), Acetyl (Protein N-term) and Phospho (STY) were added as variable modifications with a maximum number of 5 modifications per peptide. Database search was done against the human Swiss-Prot database, accessed in October 2023. Phosphorylation site intensities from the pSTY table were loaded into Perseus (version 2.0.5.0) and filtered to remove contaminants, decoy sequences and to keep only Class I phosphosites (localization probability score >0.75). Data was log2 transformed, normalized by median subtraction for each sample and subjected to statistical testing (two-tailed Student's t-test). Data was subsequently loaded into R Studio (2024.04.1 + 748) for clustering and visualization using the EnhancedVolcano, ComplexHeatmap and ggplot2 packages. Fisher tests to assess enrichment of validated and predicted PP2A substrates were performed using the database reported by Smith et al. [16]. Motif analysis was performed by using the sequence window of each phosphosite for detection of kinase motifs using regular expressions in R applying these rules:

PLK: [D/N/E]-X-[S/T]*
Aurora kinases: [K/R]-X-[S/T]*[^P] or [K/R]-X-X-[S/T]*[^P]
Cdk minimal consensus motif: [S/T]*-P

Where [] groups multiple residues for one position, * indicates the phosphorylated position, ^ before a certain residue indicates that it is forbidden for that position, and X represents any amino acid. The sequence logo of increasing phosphorylation sites against a background of all non-changing phosphorylation sites was generated using ggseqlogo package in R.

## Statistics & Reproducibility

No statistical method was used to predetermine sample size. For measurement of kinetochore proteins, 10–20 mitotic cells per condition per experiment were used. To score the percentages of mitotic cell fates and mitotic durations, 17–50 cells per conditions were analysed. For percentages of misalignments (fixed assays), 100 cells per condition per experiment. In terms of data exclusion, one of the directSLiM[LIE1]-DMSO-30' control replicates from the mass spectrometry experiment in Fig. 2 clearly behaved as an outlier after Principal Components Analysis, hence it was excluded from further downstream analysis (see Pearson correlations in source data file). Except for this condition ($n = 2$), all other conditions had three biological replicates ($n = 3$). The investigators were not blinded to group allocation during the collection of the data. To collect immunofluorescence images, the investigators manually selected mitotic cells expressing high levels of FLAG-PPP2R1A, as stated above. Analysis of mitotic cell fate was performed by manually selecting cells that entered mitosis during the timelapse and prior to inspection of their cell fate after mitotic entry, so biases are unlikely. Analysis of misalignment timings was performed by manually selecting cells that were arrested in metaphase prior to DMSO/Rapamycin treatment and prior to inspection of their misalignment timings, so biases are unlikely. Randomization was not relevant, since this study was performed by using cell lines expressing different constructs. Violin plots were produced using PlotsOfData - https://huygens.science.uva.nl/PlotsOfData/[56]. This allows the spread of data to be accurately visualised along with the 95% confidence intervals (thick vertical bars) calculated around the median (thin horizontal lines). This representation allows the statistical comparison between all treatments and timepoints because when the vertical bar of one condition does not overlap with one in another condition the difference between the medians is statistically significant ($p < 0.05$). Heatmaps showing mitotic cell fates after nuclear envelope breakdown and bar plots showing the frequencies of misalignment events were generated with Graphpad Prism 7.

## Reporting summary

Further information on research design is available in the Nature Portfolio Reporting Summary linked to this article.

## Data availability

All raw data from this study are available within the article, its Supplementary Figs., its Supplementary Data or on an online repository. All uncropped western blots and the raw data from all immunofluorescence quantification and live-cell imaging experiments are provided as a Source Data file. Phosphoproteomic data is presented in Supplementary Data 1-3 and has also been deposited in ProteomeXchange Consortium via the PRIDE partner repository with the dataset identifier PXD056390. Source data are provided with this paper.

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

## Acknowledgements

We would like to thank the MRC PPU cloning service, MRC PPU sequencing service, and the light microscopy facility at Dundee. We also thank Stephen Taylor, Geert Kops and Matthieu Bollen for reagents. This work was supported by a Wellcome Investigator grant to A.T.S. that funds L.A., A.C. and J.M.V. (222494/Z/21/Z). T.L. was supported by a Wellcome Trust and Royal Society Sir Henry Dale Fellowship (206211/Z/17/Z).

## Author contributions

A.T.S. conceived the project and supervised the study. L.A.A. developed the directSLiMs system with help from A.C. J.M.V. performed the MS experiments and analysis. A.C. performed the phenotypic chromosome alignment and segregation experiments. R.T. performed all cloning. T.L. co-supervised J.M.V. for the MS studies. A.T.S. wrote the initial draft of the paper, and all other authors generated figures and edited the paper.

## Competing interests

The authors declare no competing interests.
