## [Transparent Peer Review file · Nature Communications]

A chemical-genetic system to rapidly inhibit the PP2A-B56 phosphatase reveals a role at metaphase kinetochores

Corresponding Author: Professor Adrian Saurin

Version 0:

Reviewer comments:

Reviewer #1

(Remarks to the Author)

Dephosphorylation of key cell cycle regulators mediated by members of the PP2A family of serine/threonine phosphatases is critical for many cellular processes including normal cell cycle progression. Dephosphorylation of distinct cellular targets is achieved by a range of different PP2A holoenzymes, generally containing common catalytic and scaffolding subunits, but different regulatory subunits. Although good inhibitors of PP2A are available, these act by binding to and inhibiting the catalytic subunit, making specific inhibition of particular PP2A enzyme complexes impossible.

The B56 regulatory subunit is known to associate with its substrates by binding to a short linear LxxIxE motif, and Allan and colleagues now describe an interesting approach to inhibit PP2A complexes of the PP2A-B56 family by building on the previously described idea of expressing a high affinity LXXIxE-peptide to block the B56 LxxIxE-binding pocket. This approach was improved by incorporating a previously lacking element of temporal control to the inhibition. This was achieved by using rapamycin-mediated recruitment of a cytosolically expressed B56-blocking peptide fused to FRB to a PP2A scaffolding subunit FKBP-FLAG-PPP2R1A fusion.

While the overall approach described in the manuscript is interesting, there seem to be some caveats to the technology which limit the usefulness of the rapamycin-mediated B56 substrate-binding pocket blocking described here. In particular, it is problematic that the cytosolic expression of a high affinity blocking peptide appears to dampen down phosphatase activity even in the absence of rapamycin. This issue is acknowledged and to some extent addressed in the manuscript but as a result the paper reads more like a method development report. This is exacerbated by the fact that the discovery element of the manuscript is quite small as all of the B56 substrates that are discussed and confirmed have already been described. There are also a number of experimental issues which are listed below.

Key issues:

- The manuscript could be shortened significantly by focusing from the start on direct- SLIM cells with inhibiting peptides that do not lead to basal inhibition of PP2A-B56 and do not show altered cell cycle progression in the absence of rapamycin.
- The approach is not completely novel. The Nilsson lab developed a similar idea in 2020 (Kruse et al., 2020; high affinity LIXXE-tetrapeptide expression) and used this to identify PP2A-B56 substrates and analyse preferred substrate motifs. Allan and colleagues here optimised the original approach by introducing the rapamycin-induced recruitment of the inhibitory peptide to the PP2A phosphatase complex. This is a nice idea but on the whole, although the Nilsson approach did not have the same element of temporal control, the results of the mass spec screens for PP2A-B56 substrates seem broadly similar. It would therefore be interesting to carry out a careful comparison of the substrates identified by the two different methodologies. Also, a supplemental table of the peptides identified by Allan and colleagues would be useful.
- Throughout the paper BubR1-pT620 is used as a read-out for PP2A-B56 inhibition but total levels of BubR1 are not measured in the same cells. Total BubR1 is expected to rise significantly because of increased KNL1-MELT phosphorylation when PP2A-B56 is inhibited or depleted (Nijenhuis et al., 2014). Normalising the BubR1-pT620 signal against CENP-C is therefore not appropriate; BubR1-pT620 staining should be normalised against total BubR1 instead, and in figures, the staining for both total BubR1 and BubR1-pT620 in the same cell should be shown. Since there are good commercial mouse-anti-BubR1 antibodies available, co-staining with polyclonal phospho-specific antibodies is easily achievable. These controls are important here since BubR1-pT620 is a CDK1 site (i.e. the threonine is followed by a proline), and both the Kruse 2020 PP2A-B56 substrate screen and the author's own screen indicate that PP2A-B56 deselects for proline-directed phosphorylation sites. It is therefore important to exclude that BubR1-pT620 is simply increased because total levels of BubR1 are increased, not because it is a direct target of PP2A-B56.

Minor comments:

In Figure 1C, the immunoprecipitation experiment on the right-hand side lacks a Western blot for FLAG, to demonstrate that equal amounts of FLAG-PPP2R1A were pulled down in both the LIE1 and LIE1 + Rapamycin lane.

Reviewer #2

(Remarks to the Author)

Allan et al. developed a nice chemical genetic tool, directSLiMs, for investigating PP2A-B56 substrates, focusing on mitosis and kinetochore-microtubule attachment. The study was, in general, meticulously carried out. The basal inhibition of FRB-SLiMs was carefully examined, and sensible choices were made for suitable SLiMs in the directSLiMs. The manuscript demonstrated the advantages of short response windows of using direct SLiMs in investigating B56-mediated signaling. The manuscript is suitable for publication in Nature Communications after addressing the following concerns.

1. The most critical concern is how much drug-induced directSLiMs represents the endogenous level of PP2A-B56-mediated responses. The phosphorylation levels of B56 substrates could be altered by more than five-fold if the SLiM sequences in the substrates were mutated. The phospho-proteomic studies in the manuscript looked for 1.5-fold changes. Why not a more stringent cut-off?
2. What fraction of cellular PP2A incorporated FKBP-scaffold subunit fusion protein is unclear. Addressing this question is crucial because it would tell what fraction of PP2A-B56 holoenzymes could not respond to the compound.

Minor comments:

- It would be helpful to show the basal effects of rapamycin and rapalog in the absence of directSLiMs, particularly for global phospho-proteomic studies.

Reviewer #3

(Remarks to the Author)

In this study, La and colleagues investigate the substrate specificity of PP2A-B56 phosphatases in mitosis using proteomics and some clever cell biology. The authors create a rapamycin/rapalog regulated fusion between a PP2A-B56 docking consensus motif and the C-terminus of the phosphatase scaffold subunit. This allows them to control the interaction of PP2A-B56 with candidate substrates within cells, and then analyse changes to phosphorylation using phosphoproteomics. A major part of the study is the development of the methodology and demonstration that it works effectively. The authors find that the initial high affinity binding motif used has to be modified to reduce basal inhibition of PP2A-B56 in the absence of rapamycin/rapalog. Having established the improved system, the authors then use it to show that PP2A-B56 is needed to maintain the metaphase spindle. They conclude that PP2A-B56 regulates kinetochore microtubule attachment, however this is inferred from chromosome misalignment and not directly tested directly.

Overall the data is clearly presented and described and of high quality. I do have a few suggestions.

1. Can the authors link the phosphoproteomic data with the result in Figure 6 more clearly? Which substrate(s) mediate the effect on kinetochore-microtubule attachment?
2. How many independent repeat proteomics experiments were performed (figures 3 and 5)? The text does not make it clear. It looks like one in each case. This is probably okay for Figure 3, but Figure 5 needs replicates to support a robust conclusion.
3. There are no supplementary data tables for the proteomics and it is not clear if the raw data has been uploaded to a repository such as PRIDE for future analysis. Either one of these options should be followed.

Version 1:

Reviewer comments:

Reviewer #1

(Remarks to the Author)

My concerns have largely been addressed by the authors in the revised manuscript, and I am content for this manuscript to be published.

Reviewer #2

(Remarks to the Author)

The revised manuscript has addressed the majority of the concerns. One remaining concern is related to the basal effects of rapamycin and rapalog in the absence of directSLiMs for global phospho-proteomic studies. The authors claim a lack of effect of either inhibitor after 30 mins treatment.

As a well-known inhibitor for mTOR, one of the most important kinases in cells, it is hard to imagine that rapamycin does not have any effect on the cell's phospho-proteomics. More than 100 mTor substrates were reported in phosphosite-Plus. Have the cells been engineered such that the mTOR function is no longer affected by rapamycin?

Reviewer #3

(Remarks to the Author)

The authors have made a number of changes to the manuscript in response to the comments of the different reviewers. The provision of the mass spectrometry data and some additional discussion are helpful improvements. In terms of additional functional data, the work is also improved. The authors mention NDC80 phosphorylation in the rebuttal, which is interesting and I agree likely to be important. However, my question was not specifically about NDC80 but more general in terms of what other substrates/sites that they have identified might be functionally relevant for future work. A few words could be added in the discussion, but I leave this up to the authors.

General comments to all reviewers

We thank all the reviewers for their time and effort assessing our manuscript and making valuable suggestions. We have addressed all of these points as detailed in our point-by-point responses below. In summary, our new revised manuscript now contains:

- New experiments to pinpoint Aurora B in the kinetochore microtubule detachment phenotypes after PP2A-B56 inhibition at metaphase (Figure 6E-I, Supplementary Figure 6B-C, and Supplementary Movies 7-10).
- New experiments to examine the efficiency of incorporation of FKBP-R1A into PP2A-B56 complexes (Supplementary Figure 1A)
- New experiments to compare BUBR1 total and phospho-changes after PP2A-B56 inhibition (Figure 1E and Supplementary Figure 1B-C)
- New experiments to compare directSliM PP2A-B56 inhibition with genetic inhibition of B56 (Supplementary Figure 7)
- New in-depth analysis and reporting of the proteomic dataset, including comparisons to previous Kruse et al 2020 study (Supplementary Tables 1-3)
- New discussion sections to expand on key points raised by the reviewers.

Reviewer #1 (Remarks to the Author):

Dephosphorylation of key cell cycle regulators mediated by members of the PP2A family of serine/threonine phosphatases is critical for many cellular processes including normal cell cycle progression. Dephosphorylation of distinct cellular targets is achieved by a range of different PP2A holoenzymes, generally containing common catalytic and scaffolding subunits, but different regulatory subunits. Although good inhibitors of PP2A are available, these act by binding to and inhibiting the catalytic subunit, making specific inhibition of particular PP2A enzyme complexes impossible.

The B56 regulatory subunit is known to associate with its substrates by binding to a short linear LxxIxE motif, and Allan and colleagues now describe an interesting approach to inhibit PP2A complexes of the PP2A-B56 family by building on the previously described idea of expressing a high affinity LXXIxE-peptide to block the B56 LxxIxE-binding pocket. This approach was improved by incorporating a previously lacking element of temporal control to the inhibition. This was achieved by using rapamycin-mediated recruitment of a cytosolically expressed B56-blocking peptide fused to FRB to a PP2A scaffolding subunit FKBP-FLAG-PPP2R1A fusion.

While the overall approach described in the manuscript is interesting, there seem to be some caveats to the technology which limit the usefulness of the rapamycin-mediated B56 substrate-binding pocket blocking described here. In particular, it is problematic that the cytosolic expression of a high affinity blocking peptide appears to dampen down phosphatase activity even in the absence of rapamycin. This issue is acknowledged and to some extent addressed in the manuscript but as a result the paper reads more like a method development report. This is exacerbated by the fact that the discovery element of the manuscript is quite small as all of the B56 substrates that are discussed and confirmed have already been described. There are also a number of experimental issues which are listed below.

Author response: We thank the reviewer for carefully considering our manuscript and we would like to make the following general points about their overall assessment.

- The manuscript is indeed partly a method development study, but we believe that a novel method to temporally control PP2A-B56 activity will be crucial to characterise the substrates and processes that this enzyme controls. In our opinion, an inhibitory system that works within seconds, compared to the previous best strategy of 12 h, is a big step forward.

- Whilst we did encounter issues with basal inhibition in the absence of drug, the experimental approaches we took to resolve these problems are very informative, and likely cross applicable when applying directSLiMs to inhibit other enzymes/proteins. This should facilitate future application of this method, which we believe could be a simple yet powerful tool to manipulate protein-protein interactions and inhibit other "undruggable" enzymes/proteins.

- We used this method to make the important discovery that PP2A-B56 is needed to maintain attachments at metaphase. We have now expanded this data at revision to reveal a role for PP2A-B56 in balancing Aurora B activity and allowing tension-sensing at metaphase kinetochores. This is a significant breakthrough because previous models of tension-sensing focussed mainly on kinases. This serves to highlight the value of our system to rapidly inhibit PP2A-B56 because it would not have been possible to implicate this phosphatase in tension-sensing without such a system. This provides the rationale for others to use our system to study PP2A-B56 in the many other situations where rapid inhibition will also be crucial.

Key issues:

- The manuscript could be shortened significantly by focusing from the start on direct- SLiM cells with inhibiting peptides that do not lead to basal inhibition of PP2A-B56 and do not show altered cell cycle progression in the absence of rapamycin.

Author response: We feel that it is important to emphasize the evolution of the method to allow others to correctly apply directSLiMs to inhibit other phosphatase, enzymes or proteins. Basal inhibition will be a common feature that needs to be optimised empirically each time. In addition, there are data in the early figures that are crucial for interpretation of the later figures. For example, it is important for the reader to see that rapamycin or rapalog have little effect on basal phosphorylation in the absence of directSLiMs (Figures 2B-C). As mentioned by reviewer 2, this is a crucial experiment and it was only because of this earlier data that we were able to switch the controls in later experiments to include the equally important rapamycin or rapalog in directSLiM-AAA cells.

- The approach is not completely novel. The Nilsson lab developed a similar idea in 2020 (Kruse et al., 2020; high affinity LIXE-tetrapeptide expression) and used this to identify PP2A-B56 substrates and analyse preferred substrate motifs. Allan and colleagues here optimised the original approach by introducing the rapamycin-induced recruitment of the inhibitory peptide to the PP2A phosphatase complex. This is a nice idea but on the whole, although the Nilsson approach did not have the same element of temporal control, the results of the mass spec screens for PP2A-B56 substrates seem broadly similar. It would therefore be interesting to carry out a careful comparison of the substrates identified by the two different methodologies. Also, a supplemental table of the peptides identified by Allan and colleagues would be useful.

Author response: We agree it is an evolution of the previous approach to introduce a temporal aspect to the inhibition, but we would like to stress that this is a crucial upgrade because without that temporal control it is very difficult to study phosphatases. We can now examine acute temporal phospho-changes and study the role of PP2A-B56 in new cell cycle phases.

We have now added supplementary tables 1-3 to display details of the identified peptides and to allow simple comparison of our data to that of the Kruse et al study. Whilst there is some overlap between the studies, there are also many differences. We find 513 phosphorylation sites that increased >1.5 fold in one of our datasets, and of these 33 had been identified previously by Kruse et al (6.4%): see new supplementary table 3. We found a total of 392 unique proteins on which at least 1 phosphorylation site increased >1.5 fold in one of our datasets, and 75 of these proteins had been identified previously by Kruse et al (19.1%), albeit with a different phosphorylation site. Together, this shows there was good correlation between the studies, as expected, but our dataset clearly identifies many new substrates of PP2A-B56. Also, since we can sample much more quickly after PP2A-B56 inhibition (30 min compared to 12 h in Kruse et al), it is more likely that the changes we observe will be direct effects of PP2A inhibition. A section comparing the studies has now been added to the discussion (lines 256-273).

- Throughout the paper BubR1-pT620 is used as a read-out for PP2A-B56 inhibition but total levels of BubR1 are not measured in the same cells. Total BubR1 is expected to rise significantly because of increased KNL1-MELT phosphorylation when PP2A-B56 is inhibited or depleted (Nijenhuis et al., 2014). Normalising the BubR1-pT620 signal against CENP-C is therefore not appropriate; BubR1-pT620 staining should be normalised against total BubR1 instead, and in figures, the staining for both total BubR1 and BubR1-pT620 in the same cell should be shown.

Since there are good commercial mouse-anti-BubR1 antibodies available, co-staining with polyclonal phospho-specific antibodies is easily achievable. These controls are important here since BubR1-pT620 is a CDK1 site (i.e. the threonine is followed by a proline), and both the Kruse 2020 PP2A-B56 substrate screen and the author's own screen indicate that PP2A-B56 deselects for proline-directed phosphorylation sites. It is therefore important to exclude that BubR1-pT620 is simply increased because total levels of BubR1 are increased, not because it is a direct target of PP2A-B56.

Author response: We thank the reviewer for raising this point. We have now added total BUBR1 data to Figure 1E, which demonstrates a very modest increase in kinetochore BUBR1, in comparison to the approximately 10-fold increase in BUBR1-pT620. We have also now repeated this analysis but without staining for FLAG-R1, so that we could co-stain BUBR1 and BUBR1-pT620 together in the same cells. This analysis is presented in Supplementary Figure 1B-C, which also shows that PP2A inhibition specifically increases BUBR1 phosphorylation at the pT620 site. Our later analysis of BUBR1-pT620 in Figure 3C, also shows the increase in BUBR1-pT620 is mainly due to increased phosphorylation and not increased BUBR1 kinetochore levels.

The point raised by the reviewer about negative selection for CDK1 sites is an important one and we now include a small section in the discussion about this (lines 264-273). Although our results and that of Kruse et al show negative selection for CDK sites, there are still CDK sites that rise upon PP2A-B56 inhibition, including the well-characterised PLK1 binding motif on BUBR1 (pT620). We speculate that localised recruitment of PP2A-B56 could be used as a mechanism to regulate these key CDK-mediated phosphorylations prior to cyclin B loss at anaphase.

Minor comments:

- In Figure 1C, the immunoprecipitation experiment on the right-hand side lacks a Western blot for FLAG, to demonstrate that equal amounts of FLAG-PPP2R1A were pulled down in both the LIE1 and LIE1 + Rapamycin lane.

Author response: The FLAG blot was actually placed on the left side of 1C, along with the probes for the other PP2A-B56 subunits. It could easily be missed by the reader though, so we have now moved this to the right side to allow direct comparison to the substrates.

Reviewer #2 (Remarks to the Author):

Allan et al. developed a nice chemical genetic tool, directSLiMs, for investigating PP2A-B56 substrates, focusing on mitosis and kinetochore-microtubule attachment. The study was, in general, meticulously carried out. The basal inhibition of FRB-SLiMs was carefully examined, and sensible choices were made for suitable SLiMs in the directSLiMs. The manuscript demonstrated the advantages of short response windows of using direct SLiMs in investigating B56-mediated signaling. The manuscript is suitable for publication in Nature Communications after addressing the following concerns.

- The most critical concern is how much drug-induced directSLiMs represents the endogenous level of PP2A-B56-mediated responses. The phosphorylation levels of B56 substrates could be altered by more than five-fold if the SLiM sequences in the substrates were mutated. The phospho-proteomic studies in the manuscript looked for 1.5-fold changes. Why not a more stringent cut-off?

Author response: We thank the reviewer for raising this important point which we have now addressed with new experiments and a new section in the discussion. We directly compared the increase in phospho-signal observed on the BUB complex after global PP2A-B56 inhibition and direct PP2A-B56 inhibition at BUBR1 (via SLiM mutation). This data is now included in Supplementary Figure 7 which shows that the increase in phospho-signals are comparable in the two situations, therefore we conclude that the drug-induced directSLiM approach does induce penetrant PP2A-B56 inhibition

The question of why most substrates increase by just a few fold is a very important one that we now address in the discussion (lines 274-289). In short, we believe that this reflects a role of PP2A-B56 is continually dephosphorylating motifs that are already phosphorylated. This could produce dynamic phosphorylation sites that are important for regulating specific responses, such as rapid switch like responses at the kinetochore.

- What fraction of cellular PP2A incorporated FKBP-scaffold subunit fusion protein is unclear. Addressing this question is crucial because it would tell what fraction of PP2A-B56 holoenzymes could not respond to the compound.

Author response: We believe that most of the PP2A had likely incorporated into the FKBP-scaffold because the blot from Figure 1C showed that the endogenous R1A disappears after knockdown and is replaced by a similar amount of FKBP-R1A. However, we have now addressed this question directly by immunoprecipitating two different B56 isoforms and probing the pulldown for the R1A protein. As shown in the new Supplementary Figure 1A, all of the R1A that is incorporated into the complex is FKBP-tagged since none of the endogenous R1A is visible. This confirms that the majority of PP2A-B56 complexes incorporate the FKBP-tagged scaffold subunit.

Minor comments:

- It would be helpful to show the basal effects of rapamycin and rapalog in the absence of directSLiMs, particularly for global phospho-proteomic studies.

Author response: We agree that this is an important control, which we chose to use in the initial proteomic experiment in Figure 2B-C. Due to the lack of effect of either inhibitor after 30 mins treatment, we then performed the later proteomic experiments using rapamycin/rapalog in directSLiM-LIE-AAA cells. This is also an important control, but we could only use this after verifying in the initial experiment that these drugs had little effect on their own.

Reviewer #3 (Remarks to the Author):

In this study, La and colleagues investigate the substrate specificity of PP2A-B56 phosphatases in mitosis using proteomics and some clever cell biology. The authors create a rapamycin/rapalog regulated fusion between a PP2A-B56 docking consensus motif and the C-terminus of the phosphatase scaffold subunit. This allows them to control the interaction of PP2A-B56 with candidate substrates within cells, and then analyse changes to phosphorylation using phosphoproteomics. A major part of the study is the development of the methodology and demonstration that it works effectively. The authors find that the initial high affinity binding motif used has to be modified to reduce basal inhibition of PP2A-B56 in the absence of

rapamycin/rapalog. Having established the improved system, the authors then use it to show that PP2A-B56 is needed to maintain the metaphase spindle. They conclude that PP2A-B56 regulates kinetochore microtubule attachment, however this is inferred from chromosome misalignment and not directly tested directly.

Overall the data is clearly presented and described and of high quality. I do have a few suggestions.

- Can the authors link the phosphoproteomic data with the result in Figure 6 more clearly? Which substrate(s) mediate the effect on kinetochore-microtubule attachment?

Author response: We have now performed additional experiments to address this point. We show that detachment is dependent on Aurora B activity because the effects of PP2A-B56 inhibition are completely rescued by combined inhibition of Aurora B (Figure 6E-I, Supplementary Figure 6 and Movies 7-10). This shows that PP2A-B56 is needed to balance Aurora B activity at metaphase, implying that PP2A-B56 activity is needed to allow kinetochores to respond properly to the changes in Aurora B activity that occur upon tension. This is a significant new finding because previous models of tension-sensing were focussed mainly on the kinase Aurora B. With respect to which substrates are directly responsible for detaching kinetochore-microtubules, we hypothesise that these are the well-established Aurora B targets at the kinetochore-microtubule interface, such as NDC80. Unfortunately, it is not possible to rescue these effects by mutations because firstly there are multiple Aurora B sites on the kinetochore-microtubule interface, and secondly, even if it were possible to mutate all of these, this would prevent error correction and impair chromosome alignment during prometaphase preventing assessment of their role in metaphase detachment.

- How many independent repeat proteomics experiments were performed (figures 3 and 5)? The text does not make it clear. It looks like one in each case. This is probably okay for Figure 3, but Figure 5 needs replicates to support a robust conclusion.

Author response: 3 biological replicates were performed for each TMT experiment. This is now also clearly stated in the legend

- There are no supplementary data tables for the proteomics and it is not clear if the raw data has been uploaded to a repository such as PRIDE for future analysis. Either one of these options should be followed.

Author response: We have now included a supplementary data table to summarise the phospho-proteomic changes (Supplementary Tables 1-3). We have also uploaded all raw data to PRIDE to allow future analysis.

Response to reviewers

Reviewer #1 (Remarks to the Author):

My concerns have largely been addressed by the authors in the revised manuscript, and I am content for this manuscript to be published.

Reviewer #2 (Remarks to the Author):

The revised manuscript has addressed the majority of the concerns. One remaining concern is related to the basal effects of rapamycin and rapalog in the absence of directSLiMs for global phospho-proteomic studies. The authors claim a lack of effect of either inhibitor after 30 mins treatment.

As a well-known inhibitor for mTOR, one of the most important kinases in cells, it is hard to imagine that rapamycin does not have any effect on the cell's phospho-proteomics. More than 100 mTor substrates were reported in phosphosite-Plus. Have the cells been engineered such that the mTOR function is no longer affected by rapamycin?

Author response: *There are two things that we would like to point out that we have clarified in the final manuscript text. Firstly, the rapalog compound only binds to the FRB-T2098L mutant, which is indeed engineered into our inhibitory system but absent from the endogenous mTOR complex. So, it is fully expected that rapalog will not inhibit the endogenous mTOR complex, thus explaining the lack of effect in Figure 2a and Supplementary Figure 2a. Secondly, rapamycin does inhibit the TORC1 complex, but we observed almost no phospho-proteomic changes after a 30-min treatment in mitotically-arrested cells (Figure 2A and Supplementary Figure 2a). In fact, it was indistinguishable from rapalog in these assays. We feel this is most likely explained by the fact that TORC1 is well-known to be rapidly inhibited by CDK1 during mitosis, as demonstrated in the following articles:*

Odle et al, Mol Cell, 2020. <https://doi.org/10.1016/j.molcel.2019.10.016>

Moustafa-Kamal et al, Cell Rep, 2020. <https://doi.org/10.1016/j.celrep.2020.108230>

Our assays were performed after a few hours of a mitotic arrest. Therefore, we anticipate that TORC1 is fully inhibited and its substrates are already dephosphorylated at this timepoint, hence explaining the lack of additional dephosphorylation following rapamycin treatment. We have emphasized these key points and cited the relevant studies in the revised manuscript.

Reviewer #3 (Remarks to the Author):

The authors have made a number of changes to the manuscript in response to the comments of the different reviewers. The provision of the mass spectrometry data and some additional discussion are helpful improvements. In terms of additional functional data, the work is also improved. The authors mention NDC80 phosphorylation in the rebuttal, which is interesting and I agree likely to be important. However, my question was not specifically about NDC80 but more general in terms of what other substrates/sites that they have identified might be functionally relevant for future work. A few words could be added in the

discussion, but I leave this up to the authors.

Author response: *It is possible that Aurora B itself is regulated by PP2A-B56, since we see increases in Aurora B and Borealin phosphorylation after PP2A-B56 inhibition. We feel it is important to test this and the possibility that PP2A-B56 specifically regulates Aurora B targets, such as NDC80. These ideas are covered in the discussion paragraph from Line 310.*